# Learning Efficient Fuse-and-Refine for Feed-Forward 3D Gaussian Splatting

**Yiming Wang**
ETH Zurich

**Lucy Chai**
Google

**Xuan Luo**
Google

**Michael Niemeyer**
Google

**Manuel Lagunas**
Google

**Stephen Lombardi**
Google

**Siyu Tang**
ETH Zurich

**Tiancheng Sun**
Google

## Abstract

Recent advances in feed-forward 3D Gaussian Splatting have led to rapid improvements in efficient scene reconstruction from sparse views. However, most existing approaches construct Gaussian primitives directly aligned with the pixels in one or more of the input images. This leads to redundancies in the representation when input views overlap and constrains the position of the primitives to lie along the input rays without full flexibility in 3D space. Moreover, these pixel-aligned approaches do not naturally generalize to dynamic scenes, where effectively leveraging temporal information requires resolving both redundant and newly appearing content across frames. To address these limitations, we introduce a novel Fuse-and-Refine module that enhances existing feed-forward models by merging and refining the primitives in a canonical 3D space. At the core of our method is an efficient hybrid Splat-Voxel representation – from an initial set of pixel-aligned Gaussian primitives, we aggregate local features into a coarse-to-fine voxel hierarchy, and then use a sparse voxel transformer to process these voxel features and generate refined Gaussian primitives. By fusing and refining an arbitrary number of inputs into a consistent set of primitives, our representation effectively reduces redundancy and naturally adapts to temporal frames, enabling history-aware online reconstruction of dynamic scenes. Trained on large-scale static scene datasets, our model learns an effective global strategy to process around 200k primitives within 15ms and significantly enhances reconstruction quality compared to pixel-aligned reconstruction approaches. Without additional training, our model generalizes to video by fusing primitives across time, yielding a more temporally coherent result compared to baseline methods with graceful handling of occluded content. Our approach achieves state-of-the-art performance in both static and streaming scene reconstructions while running at interactive rates (15 fps with 350ms delay) on a single H100 GPU.

## 1  Introduction

Novel view synthesis from sparse views is a core problem for augmented and virtual reality (AR/VR) applications, where practical constraints require the reconstruction to be fast and accurate for a positive user experience. Recent feed-forward 3D Gaussian Splatting methods [78, 4, 84] trained with large-scale scene datasets offer efficient solutions for the sparse-view reconstruction problem. The key idea of these approaches is to learn a large feed-forward network, typically Transformer-based, to predict Gaussian primitives and use splatting as a fast renderer [22].

The common approach taken by these methods is to regress per-pixel Gaussian primitives that lie along the input camera rays. However, this leads to redundancy in the representation when input

39th Conference on Neural Information Processing Systems (NeurIPS 2025).

views have overlapping content, and can also lead to incomplete geometry with occluded content not visible from the input views. This per-pixel representation also does not naturally generalize to dynamic scenes, where redundant primitives between frames should be merged and new primitives should emerge to capture newly appearing content.

To address these limitations, we introduce a novel Fuse-and-Refine module that processes and merges Gaussian primitives in a canonical 3D space. Our approach enables more flexible primitive placement and better density control, leading to improved reconstruction quality in static scenes. It also greatly facilitates streaming reconstruction in dynamic scenes by avoiding error accumulation and redundant primitives over time, while better handling temporal occlusions and disocclusions through the use of historical information. We note that past approaches to merging primitives have relied on heuristics [23] that typically degrade reconstruction quality and require additional time-consuming gradient descent optimization steps. In contrast, our learning-based solution leverages a feed-forward Transformer trained on large-scale scene datasets combined with the structured nature of voxels as an intermediate scaffold to efficiently fuse and refine Gaussian primitives.

We design a hybrid Splat-Voxel model that proceeds as follows: given a set of initial Gaussian primitives from existing pixel-aligned feed-forward models, we first perform a Splat-to-Voxel Transfer to aggregate local features from Gaussian primitives into a voxel representation, and then apply a Transformer to generate refined Gaussian primitives from voxel features. Our voxel representation is specifically designed to ensure both efficiency and high-quality output. We use a coarse-to-fine voxel hierarchy to balance detail-preservation and computational overhead, operating a sparse voxel transformer on features at the coarse resolution while refining the primitives at a high resolution. Our Fuse-and-Refine module can merge 200K Gaussian primitives from existing feed-forward 3D Gaussian models in just 15 ms, while delivering a PSNR improvement of around 2 dB.

We train our model exclusively on static datasets, but our model also enables history-aware streaming reconstruction of sparse-view dynamic scenes without additional training or modifications. This is advantageous as multiview video datasets remain limited in both size and diversity, prohibiting large-scale training. For dynamic scenes, we first warp primitives from a past frame to the current frame, filtering out those with large warping errors, and deposit primitives from both the past and current frame onto our Splat-Voxel representation. The resulting refined primitives significantly improve the temporal consistency of existing feed-forward models, while avoiding redundant primitives and mitigating error accumulation. Our method achieves state-of-the-art performance on several multi-view video datasets under the sparse camera setup, and can reach interactive rates (15fps with 350ms delay) under a single H100 GPU.

In summary, the main contributions of our paper are as follows:

- We introduce a novel Fuse-and-Refine module that enhances existing feed-forward 3D Gaussian Splatting methods and extends them to history-aware streaming reconstruction in a zero-shot manner.
- We propose a hybrid Splat-Voxel representation that combines a coarse-to-fine voxel hierarchy with a sparse voxel Transformer, enabling effective learning of the Fuse-and-Refine module on large-scale static scene datasets.
- Extensive experimental results demonstrate that our approach achieves state-of-the-art performance on both static and streaming-based dynamic scene reconstruction, while maintaining interactive runtimes.

## 2 Related Work

**Novel View Synthesis** A wide range of methods have been developed over the past decades for novel view synthesis (NVS). A key design decision is the choice of 3D scene representation to establish correspondences and capture structure. Popular representations include multi-plane images [16, 14, 3, 72, 66, 73, 30], neural fields [41, 57], voxel grids [62, 42, 50], and point clouds [77, 22]. In this work, we adopt 3D Gaussian Splatting [22] as our final rendering representation due to its ability to achieve high-quality and real-time consistent novel view synthesis.

Existing NVS methods can be broadly divided into those that optimize scene representation at test time and those that directly predict it through a feed-forward network [68, 60, 53, 82, 13, 21, 55, 54, 20]. The combination of Transformers and 3D Gaussian Splatting has emerged as an efficient approach to

predict pixel-aligned Gaussian splatting in a data-driven, feed-forward manner [84, 6, 4, 83, 64, 71]. However, pixel-aligned prediction design often results in suboptimal scene reconstruction due to limited flexibility in accurately positioning primitives in 3D space. To address this limitation, several works have leveraged feed-forward networks operating on 3D representations, such as graph [86], voxel [39, 52, 5], or point cloud [7], to process 3D Gaussian Splatting. Yet, extending these representations to dynamic scenes remains challenging and largely unexplored. To this end, we propose a hybrid point-voxel representation that can generalize to dynamic scenes during inference while trained only on static scenes. Our design shares a similar spirit with voxel-based point cloud processing networks [88, 36, 38], which leverage voxel structures for efficient point cloud processing. However, existing methods [31] typically rely on autonomous-driving–specific projections (e.g., bird's-eye view) to handle large-scale scenes. In contrast, we construct a cost volume suitable for general unbounded scenes and introduce a coarse-to-fine voxel hierarchy that enables global attention at the coarsest level.

**Reconstructing Dynamic Scenes**   Handling dynamic motions in novel view synthesis (NVS) is challenging due to occlusions and the need for temporal consistency. Dynamic scene reconstruction is typically addressed in multi-view setups with fixed cameras [79, 58, 74, 9, 37, 40] or monocular settings where both the camera and scene move [44, 28, 75, 29, 69]. Most methods optimize on the entire video sequence, relying on learned or hand-crafted priors such as optical flow [65, 34], tracking [11], and depth estimation [45, 80, 49, 48] for supervision [28, 75]. Our method leverages pretrained models for point tracking and optical flow as motion priors and operates causally and online, using only previously received frames without requiring access to the full video.

Streaming reconstruction [9] presents a significant challenge in dynamic scene reconstruction, aiming to enable online novel view synthesis from potentially unbounded video streams for real-world applications. Several methods [70, 26, 63, 17] have explored the use of historical information to accelerate optimization, but remain computationally expensive and depend on dense multi-view inputs. Feed-forward models are well-suited for sparse multi-view streaming scene reconstruction. However, many of these methods can only take a single frame as input [15, 33, 51, 84, 4], ignoring temporal information and consequently suffering from flickering and occlusion artifacts. Moreover, the scarcity of real-world multi-view dynamic datasets hinders the ability to train feed-forward dynamic models [51, 32] for real-world streaming applications. By generalizing from static scene training to dynamic video inference without requiring additional temporal data, our method enables efficient, history-aware novel view synthesis for streaming scenarios.

## 3   Preliminaries: Feed-forward 3D Gaussian Splatting

**3D Gaussian Splatting (3DGS) [22]**   3DGS represents a scene as a set of anisotropic 3D Gaussian primitives, each defined by its position $\boldsymbol{\mu}_i \in \mathbb{R}^3$, opacity $\alpha_i \in \mathbb{R}$, quaternion rotation $\mathbf{r}_i \in \mathbb{R}^4$, scale $\mathbf{s}_i \in \mathbb{R}^3$, and spherical harmonic coefficients $\mathbf{c}_i \in \mathbb{R}^S$, where $S$ denotes the number of coefficients used for modeling view-dependent color. The rendered color at pixel $\mathbf{p}$, denoted as $C(\mathbf{p})$, is computed via splatting-based rasterization, where each 3D Gaussian is projected to 2D screen space and blended using front-to-back alpha compositing:

$$C(\mathbf{p}) = \sum_{i=1}^{K} \mathbf{c}_i \alpha_i \mathcal{G}_i^{\text{2D}}(\mathbf{p}) \prod_{j=1}^{i-1} \left(1 - \alpha_j \mathcal{G}_j^{\text{2D}}(\mathbf{p})\right) \tag{1}$$

Here, $\mathcal{G}_i^{\text{2D}}(\mathbf{p})$ denotes the 2D projection of the $i$-th 3D Gaussian at pixel $\mathbf{p}$.

**Feed-forward Reconstruction Model**   To predict the 3D Gaussian primitives in a feed-forward manner from sparse-view images, we adopt the multi-view Transformer architecture from GS-LRM [84]. Given $N$ RGB images of resolution $H \times W$ with known camera intrinsics and extrinsics, we extract per-pixel features by concatenating RGB values with Plücker ray embeddings [46]. These features are grouped into non-overlapping $p \times p$ image patches [12] and encoded using a shallow CNN to obtain patch features of channel dimension $C$. The patch features from all $N$ views are then flattened into a sequence of $N \times \frac{HW}{p^2}$ tokens, denoted as $\{\mathbf{X}\}$. These tokens are processed by a multi-view Transformer [67] to jointly model geometric structure and appearance across views, producing latent features $\{\mathbf{Z}\}$:

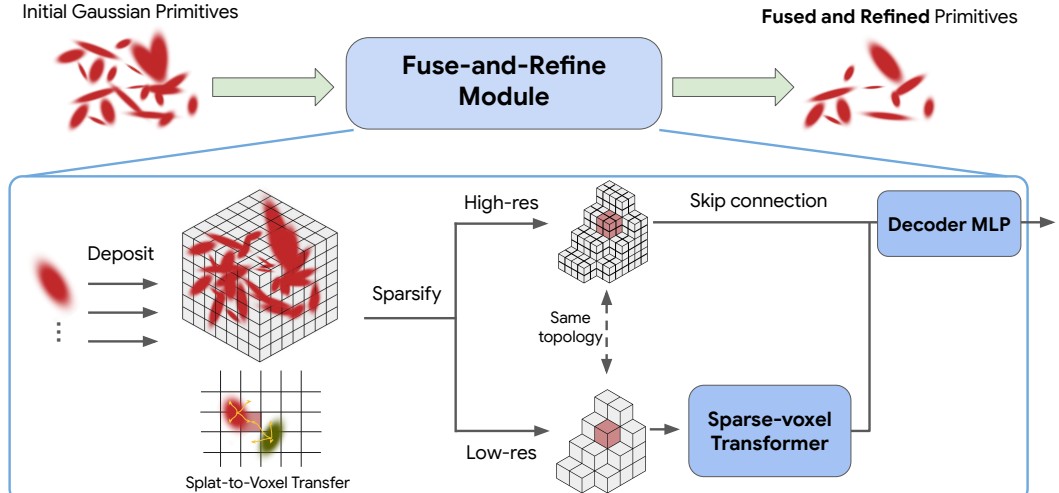

Figure 1: **Method Overview.** Our Fuse-and-Refine module takes as input Gaussian primitives produced by existing feed-forward models, either from the current frame or warped from previous reconstructions, and produces fused and refined primitives that improve scene reconstruction. The input primitives are first deposited into a high-resolution voxel grid with a splat-to-voxel Transfer strategy, which is then adaptively sparsified to construct a coarse-to-fine voxel hierarchy. A sparse voxel transformer is applied at the coarse level to capture global context, and new primitives are subsequently generated at the high-resolution level.

$$\{\mathbf{Z}\} = \text{Transformer}_{MV}(\{\mathbf{X}\}) \tag{2}$$

Finally, a linear projection is applied to each latent token to produce an initial set of Gaussian primitives and associated feature vectors:

$$\{\mathbf{G}_k, \mathbf{F}_k\} = \text{Linear}(\{\mathbf{Z}\}) \tag{3}$$

Here, $\mathbf{G}_k$ represents the parameters of a 3D Gaussian primitive aligned with the $k$-th image patch, which can be directly rendered using Gaussian Splatting [22], while $\mathbf{F}_k$ denotes an additional feature vector used for the subsequent learning-based fusion and refinement.

## 4   Learning to Fuse and Refine

Given an arbitrary set of initial Gaussian primitives, for instance those from multi-view images (as in Sec. 3) or from previous time steps (as we will describe in Sec. 5), we aim to fuse nearby primitives and produce refined primitives that improve novel view synthesis results.

**Challenges**   Preserving the fidelity of millions of Gaussian primitives while merging them is highly non-trivial, with several factors contributing to the challenge. First, maintaining the original 3D Gaussian distributions without divergence is fundamentally infeasible when combining multiple kernels. This issue is further exacerbated in the context of sparse-view novel view synthesis, where the fusion problem is inherently ill-posed. In addition to merging the Gaussian distribution determined by primitive mean and covariance, other rendering attributes like color, opacity, scale, and rotation must also be reconciled with limited supervision from sparse 2D observations. As a result, heuristic approaches [23] often lead to degraded reconstruction quality.

**Solution**   Instead, we propose an efficient learning-based approach that not only merges 3D primitives, but also further refines them through a feed-forward Transformer. Trained on large-scale scene data, our model learns a global strategy to fuse input primitives into a refined set of splats in just 15 ms, significantly improving reconstruction performance. The proposed learning framework adopts a hybrid Splat-Voxel representation, where an intermediate voxel grid serves as a structured spatial proxy for aggregating and distributing splats. A voxel Transformer operates in latent space to fuse information in a canonical 3D space and produce refined splats. An overview of the proposed Fuse-and-Refine module is illustrated in Fig. 1.

## 4.1 Splat-to-Voxel Transfer

Starting with the predicted Gaussian primitives $\mathbf{G}_k$ and their associated image features $\mathbf{F}_k$ from the multi-view Transformer (Eqn. 2), we first operate a Splat-to-Voxel Transfer procedure to convert the point-based representation to a structured voxel representation. Implementation details, including the choice of kernel function $\mathcal{K}$ and other parameters, are provided in the supplementary material (Sec. A).

**Splat Depositing**   Given a dense high-resolution voxel grid $V$, we deposit each Gaussian primitive $\mathbf{G}_k$ to the set of $\{\mathbf{M}_k\}$ of nearest voxels based on the position $\mu_k$ using a distance kernel $\mathcal{K}$. That is, for voxel $V_j$, its weight associated with $\mathbf{G}_k$ is:

$$w_{jk} = \begin{cases} \frac{\mathcal{K}(\mu_k - \mathbf{x}_j)}{\sum_{j' \in \{\mathbf{M}_k\}} \mathcal{K}(\mu_k - \mathbf{x}_{j'})}, & \text{if } j \in \{\mathbf{M}_k\}, \\ 0 & \text{otherwise.} \end{cases} \tag{4}$$

where $\mathbf{x}_j$ represents the center of the $j$-th voxel. We choose the adjacent 8 voxels for each splat in our implementation.

**Splat Fusion**   Each voxel $V_j$ then accumulates the deposited splat attributes using opacity $\alpha$ as weighting. The corresponding voxel attribute $\mathbf{E}_j$ at voxel $V_j$ is then the normalized weighted sum of the splat features $\mathbf{F}_k$ in the voxel:

$$\mathbf{E}_j = \frac{1}{\mathbf{W}_j} \sum_{k \in \{\mathbf{M}_k\}} w_{jk} \alpha_k [\mathbf{F}_k; \mathbf{G}_k], \quad \mathbf{W}_j = \sum_{k \in \{\mathbf{M}_k\}} w_{jk} \alpha_k \tag{5}$$

where $[\mathbf{F}_k; \mathbf{G}_k]$ denotes the concatenation of the feature vector $\mathbf{F}_k$ and Gaussian attributes $\mathbf{G}_k$. The interpolation weight $w_{jk}$ is computed as described in Eqn. 4. We use the notation $k \in \mathbf{M}_k$ to indicate that $k$ indexes over the set of all Gaussian primitives, each associated with a weight, Gaussian attributes, and a feature vector.

**Coarse-to-fine voxel hierarchy**   Once the splat features are deposited onto the voxels, we construct a coarse-to-fine voxel hierarchy. Specifically, the high-resolution voxels are downsampled by a factor of $[d, h, w]$ to generate a low-resolution voxel grid that serves as the coarse scene representation. Sharing the same spatial topology, each coarse-level voxel directly corresponds to a $[d, h, w]$ block of fine-level voxels and can be restored by subdivision. We denote $\mathbf{N}_i$ as the set of $d \times h \times w$ fine voxels associated with the $i$-th coarse voxel. Using the previously computed voxel features $\mathbf{E}_j$ as the fine voxel features, we now compute coarse voxel features $\mathbf{E}_i^c$.

We derive coarse voxel features using a shallow multi-layer perceptron (MLP) on the concatenation of the fine features and set coarse voxel weights as the sum of the associated fine voxel weights:

$$\mathbf{E}_i^c = \text{MLP}_{coarse} \left( [\mathbf{E}_j \mid j \in \mathbf{N}_i] \right), \quad \mathbf{W}_i = \sum_{j \in \mathbf{N}_i} \mathbf{W}_j \tag{6}$$

**Voxel Sparsification**   We leverage geometric cues from coarse voxel weights $\mathbf{W}_i$, which encapsulate the spatial distribution and opacity of Gaussian primitives within each voxel, to efficiently cull empty and insignificant regions of the scene. We sort the coarse-level voxels by their voxel weights and retain only the top 20% with the largest weights as a sparse voxel representation of the scene. This sparsification is crucial for the downstream voxel transformer, and reduces the input token count to fewer than $10K$, which allows the voxel transformer to run in 15 ms. Note that the high-resolution fine-level voxels are also sparsified accordingly if their corresponding coarse-level voxels are culled.

## 4.2 Sparse Voxel Transformer

From coarse voxel features, we treat each voxel feature as a token and reshape them to 1D vectors. We train a transformer to process the set of coarse voxel features $\mathbf{E}^c$:

$$\{\mathbf{O}^c\} = \text{Transformer}_{voxel}(\{\mathbf{E}^c\}) \tag{7}$$

The processed latent features $\{\mathbf{O}^c\}$ are then replicated from the coarse voxel grids within their corresponding fine voxel grids, producing latent features $\{\mathbf{O}_j\}$ at the fine voxel resolution. A shallow

Table 1: **Quantitative Comparisons on the RealEstate10K Dataset.** Our method achieves state-of-the-art performance among recent feed-forward 3D Gaussian Splatting methods.

| Method | PSNR↑ | SSIM↑ | LPIPS↓ |
|---|---|---|---|
| pixelSplat [4] | 26.09 | 0.863 | 0.136 |
| MVSplat [6] | 26.39 | 0.869 | 0.128 |
| TranSplat [83] | 26.69 | 0.875 | 0.125 |
| HiSplat [64] | 27.21 | 0.881 | 0.117 |
| OmniScene [71] | 26.19 | 0.865 | 0.131 |
| DepthSplat [76] | 27.47 | 0.889 | 0.114 |
| GS-LRM [84] | 28.10 | 0.892 | 0.114 |
| **Ours** | **28.47** | **0.907** | **0.078** |

Table 2: **Quantitative Comparisons on the DL3DV Dataset.** We build our Voxel Transformer on GS-LRM [35] and present ablation studies on model design. For a fair comparison, we use 12 multi-view Transformer layers with our Voxel Transformer to match the size and runtime of GS-LRM, which uses 24 layers.

| Method | PSNR↑ | SSIM↑ | LPIPS↓ | Time(ms) |
|---|---|---|---|---|
| GS-LRM | 28.59 | 0.925 | 0.063 | 52.8 |
| + Non-learning Fusion | 12.57 | 0.357 | 0.741 | 33.8 |
| + Ours (w/o Coarse-to-fine) | 20.62 | 0.587 | 0.247 | 48.6 |
| + Ours (w/o Sparse Voxel) | 29.69 | 0.926 | 0.060 | 72.0 |
| + Ours (w/ 3D CNN) | 29.44 | 0.922 | 0.061 | 82.1 |
| + Ours (w/o Splat Feature) | 29.40 | 0.924 | 0.061 | 49.2 |
| + **Ours** | **30.61** | **0.935** | **0.052** | 52.5 |

MLP subsequently generates refined Gaussian primitives by integrating the initial fine-level voxel attributes $\mathbf{E}_j$ with the fine-level transformer latents $\mathbf{O}_j$.

$$\mathbf{G}'_j = \text{MLP}_{fine}([\mathbf{O}_j, \mathbf{E}_j]) \tag{8}$$

where $\mathbf{G}'_j \in \mathbb{R}^l$ represents the predicted Gaussian primitives around the sparsified fine-level voxel grids, with $l$ denoting the number of rendering parameters.

## 4.3 Training

We train the SplatVoxel feed-forward networks on multi-view static scene datasets [87, 35] using a photometric loss combining Mean Squared Error (MSE) and perceptual LPIPS [85]:

$$\mathcal{L} = \mathcal{L}_{\text{MSE}}(\hat{I}, I) + \lambda\mathcal{L}_{\text{LPIPS}}(\hat{I}, I) \tag{9}$$

where $\hat{I}$ denotes ground-truth target image and $I$ the corresponding rendered target image from the predicted Gaussian primitives, which come from either the multi-view transformer or voxel transformer. We set $\lambda$ to 0.5 for the multi-view transformer and 4.0 for the voxel transformer. We train our full-scale model for cross-dataset generalization on the DL3DV dataset [35] with a batch size of 128 for a total of 300K iterations using a two-stage training strategy. In the first stage, we train the multi-view Transformer backbone for 200K iterations, followed by joint fine-tuning of both the multi-view and voxel Transformers for an additional 100K iterations. Network architecture and further training details are provided in the supplementary material (Sec. A.1 and Sec. B).

## 5 Zero-shot Streaming Fusion

In this section, we demonstrate an application of the Sparse Voxel Transformer, trained solely on static scenes, to enable zero-shot history-aware novel view streaming without requiring any training on dynamic scenes. Please refer to the Appendix (Sec. A.3) for full implementation details.

**3D Warping** Given a Gaussian primitive with an initial position $\mu_{t'}$ at a previous frame $t'$, we first estimate its corresponding 3D position $\mu_t$ at the current frame $t$ by performing triangulation on the 2D correspondence obtained by a pre-trained 2D point tracking model [11]. In detail, we first compute its 2D projection $\{\mathbf{p}_{t'}^i \mid i = 1...N\}$ across $N$ input views. We then run 2D point tracking on the projected points to obtain their corresponding 2D pixel position $\mathbf{p}_t^i$ in the current frame. The 3D position is subsequently recovered by finding the closest point to the $N$ camera rays originating from the 2D projections $\mathbf{p}_t^i$. We adopt embedded deformation graph [61, 43, 25] to propagate reliable motion estimates from a sparse set of anchor points to the entire scene. Anchor points are selected via farthest point sampling [47], and an embedded deformation graph is constructed by connecting each splat to its $K$ nearest anchor points, with blend weights computed based on spatial distances. For efficiency, we only maintain primitives from a set of past keyframes, and the contribution of these primitives are smoothly adjusted as older keyframes are replaced by newer ones to ensure seamless temporal transitions.

Table 3: **Quantitative Comparison Under Varying Input Views on the DL3DV Dataset.** Trained with 4 input views, our method consistently outperforms GS-LRM across different input-view settings at inference.

| Method | 2 Views | | | 4 Views | | | 8 Views | | | 16 Views | | |
|---|---|---|---|---|---|---|---|---|---|---|---|---|
| | PSNR↑ | SSIM↑ | LPIPS↓ | PSNR↑ | SSIM↑ | LPIPS↓ | PSNR↑ | SSIM↑ | LPIPS↓ | PSNR↑ | SSIM↑ | LPIPS↓ |
| GS-LRM | 18.99 | 0.810 | 0.135 | 28.59 | 0.925 | 0.063 | 22.51 | 0.879 | 0.106 | 20.12 | 0.792 | 0.177 |
| **Ours** | **26.32** | **0.876** | **0.093** | **30.34** | **0.934** | **0.054** | **28.98** | **0.930** | **0.060** | **26.06** | **0.889** | **0.093** |

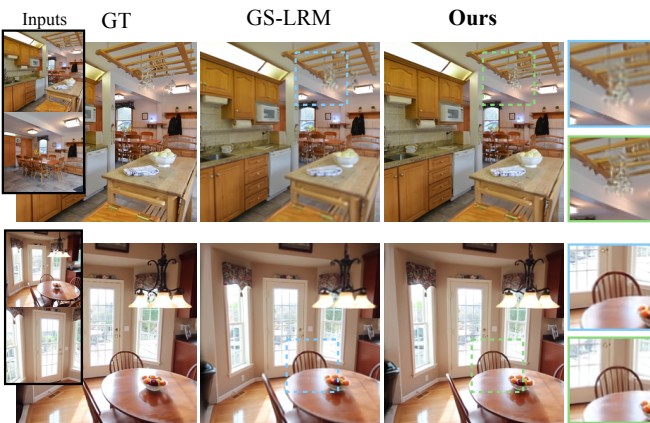

Figure 2: **Static Scene Reconstruction.** Qualitative comparison of our method with GS-LRM [84] on RealEstate10k [87]. Our method preserves high fidelity reconstructions and sharp details.

**Error-aware Fusion** During streaming reconstruction, we incorporate both historical Gaussian primitives warped from keyframes and new Gaussian primitives predicted from multi-view observations at the current frame. All of the collected Gaussian primitives are deposited to voxels (Sec. 4.1) and then processed by the voxel transformer (Sec. 4.2), producing refined, history-aware Gaussian primitives. To mitigate artifacts resulting from warping inaccuracies of historical primitives (e.g. warping error, appearance variation, or new objects), we use an adaptive fusion weighting mechanism during splat fusion based on the per-pixel error between the past and current frame splats when rendered to the input views.

# 6 Experiments

In this section, we present evaluations of feed-forward novel view synthesis on both static and dynamic scenes, along with ablation studies on our splat-voxel representation designs.

## 6.1 Feed-forward Novel View Synthesis

We benchmark our method on two widely used datasets, RealEstate10K [87] and DL3DV [35], which cover both indoor scenes and unbounded large-scale environments.

Following the training and testing protocol of [4], our method achieves state-of-the-art performance on RealEstate10K, as shown in Table 1. As illustrated in Fig. 2, our method reconstructs finer details such as thin structures (e.g., chairs) and reflections compared to the strongest baseline, GS-LRM [84]. Additional qualitative comparisons and training details are provided in the Appendix (Sec. B and Sec. D).

We then demonstrate the ability of our method to enhance existing feed-forward 3D Gaussian Splatting approaches through a comparison on the DL3DV dataset [35] with GS-LRM [84]. Table 2 demonstrates that our method outperforms GS-LRM by approximately 2 dB in PSNR, while maintaining a comparable number of network parameters, similar inference time, and the same training configuration. Our method also demonstrates strong generalization capability across varying input views. Trained with 4 input views on the DL3DV dataset, our method consistently outperforms

Table 4: **Novel View Streaming Metrics.** We compare reconstruction quality of our model to prior methods on two multi-view video datasets with sparse-view input. We report the average running time of processing a single frame on Neural3DVideo at the resolution of $320 \times 240$. The best and second-best results are highlighted for clarity.

| Method | Time (s) | Neural3DVideo [27] | | | | LongVolumetricVideo [79] | | | |
|---|---|---|---|---|---|---|---|---|---|
| | | PSNR↑ | SSIM↑ | LPIPS↓ | Flicker↓ | PSNR↑ | SSIM↑ | LPIPS↓ | Flicker↓ |
| 3DGS [22] | 9.8 | 21.02 | 0.6234 | 0.4989 | 43.51 | 18.56 | 0.6368 | 0.4334 | 61.69 |
| 3DGStream [63] | 3.8 | 14.24 | 0.3542 | 0.6016 | 6.554 | 16.97 | 0.4405 | 0.5040 | 12.44 |
| 4DGS [74] | 6.0 | 23.16 | 0.7812 | 0.2079 | 2.746 | 18.75 | 0.6361 | 0.3554 | 6.528 |
| GS-LRM [84] | 0.04 | 21.80 | 0.8488 | 0.1278 | 5.714 | 25.49 | 0.8558 | 0.1377 | 8.463 |
| **Ours** | 0.07 (non-keyframe) 0.35 (keyframe) | 27.41 | 0.8863 | 0.1040 | 2.916 | 25.56 | 0.8645 | 0.1242 | 7.782 |

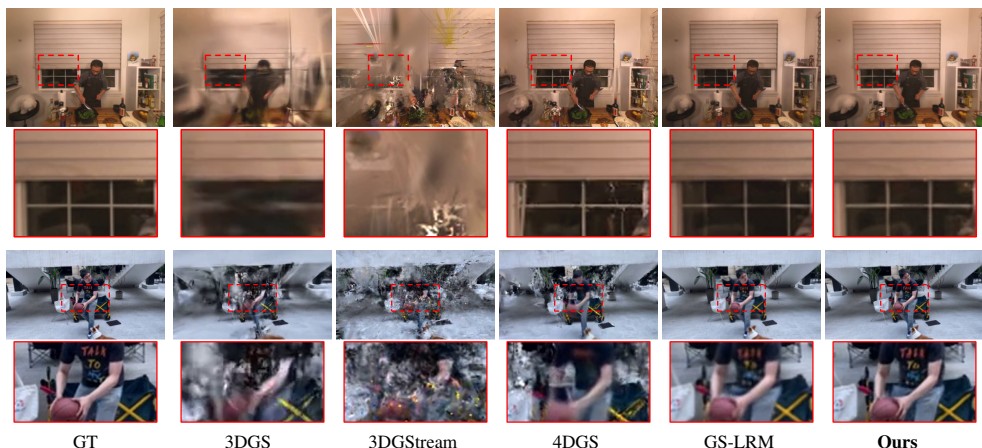

| GT | 3DGS | 3DGStream | 4DGS | GS-LRM | **Ours** |

Figure 3: **Qualitative Comparison of Novel View Streaming.** We compare against per-frame methods (3DGS and GS-LRM) and temporal methods (4DGS and 3DGStream) on LongVolumetricVideo [79] with 4 input views and Neural3DVideo [27] with 2 input views.

GS-LRM across awide range of input configurations as shown in the Table 3. These results further highlight the strong generalization ability of our method for long-sequence, large-scale scene reconstruction. A compairson with Gaussian Graph Network [86] is provided in the appendix Table 13, which state-of-the-art method for handling diverse input views.

## 6.2 Novel View Streaming

We conduct a cross-dataset evaluation of our method on dynamic scene reconstruction, demonstrating its generalization capability in enhancing reconstruction quality and improving temporal coherence. We train both GS-LRM and our model on DL3DV, and compare our method with baselines on the novel view streaming task on two dynamic scene datasets, Neural3DVideo [27] and LongVolumetricVideo [79]. We use 2 input views for Neural3DVideo and 4 views for LongVolumetricVideo. For quantitative evaluation of temporal coherence, we introduce a flicker metric that complements the per-frame novel view reconstruction scores. Temporal flicker is quantified by measuring feature-space variations over time. For predicted frames $I_t$ and ground truth $I_t^*$, we compute

$$\text{Flicker}_t = \left| D_t - D_t^* \right|,$$

where $D_t = \|\phi(I_t) - \phi(I_{t-1})\|_2$ and $D_t^* = \|\phi(I_t^*) - \phi(I_{t-1}^*)\|_2$ denote temporal feature differences extracted from a pretrained VGG network [56]. The final flicker score is computed by averaging $\text{Flicker}_t$ over the entire sequence. Lower values indicate smoother temporal consistency.

As presented in Table 4, our method achieves the highest per-frame novel view synthesis performance across all three evaluation metrics on both datasets with the second-best flicker scores. In comparison to the state-of-the-art per-frame feed-forward method GS-LRM, our approach qualitatively reconstructs geometry and appearance with greater accuracy and finer details (Fig. 6.1) Our method reduces

Table 5: **Comparison of GS-LRM with Simple 3D Warping and Fusion Strategy.** Incorporating historical information for streaming reconstruction remains challenging due to accumulated errors, whereas our Fuse-and-Refine network learns to effectively mitigate this limitation.

| Method | PSNR↑ | SSIM↑ | LPIPS↓ | Flicker↓ |
|---|---|---|---|---|
| GS-LRM | 21.80 | 0.8488 | 0.1278 | 5.714 |
| GS-LRM + 3D Warping | 21.40 | 0.8250 | 0.1419 | **2.163** |
| GS-LRM + 3D Warping + Fusion (Concat + Dropout) | 19.52 | 0.7429 | 0.2502 | 3.983 |
| GS-LRM + 3D Warping + Ours (Fuse-and-Refine) | **27.41** | **0.8863** | **0.1040** | 2.916 |

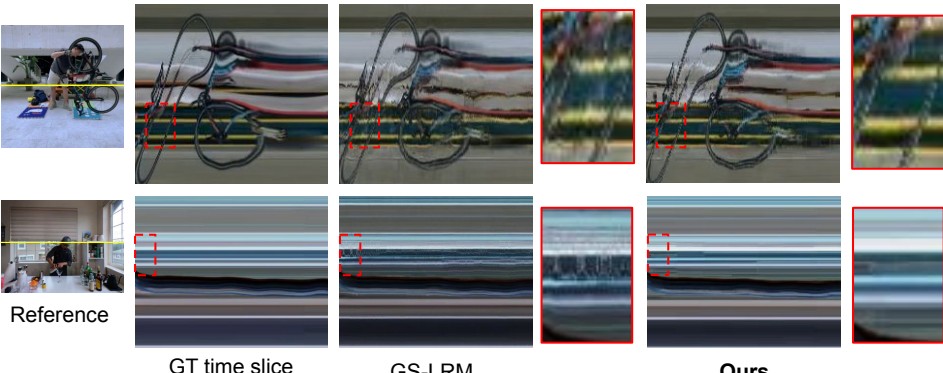

Reference      GT time slice      GS-LRM      **Ours**

Figure 4: **Temporal Slice Comparison.** We extract a horizonal line in the ground truth and generated videos to visualize temporal stability. Applying GS-LRM independently to each frame results in noticeable flickering, whereas our method achieves greater temporal consistency and improved occlusion robustness through temporal fusion and refinement.

flickering through history-aware modeling, visualized with temporal slices (Fig. 4). Although 4D Gaussian Splatting yields slightly better flicker scores, it produces significantly poorer novel views and its costly temporal optimization is impractical for online processing of long sequences.

We further apply 2D tracking and 3D warping to GS-LRM to demonstrate that directly warping past reconstructions introduces significant errors. As shown in Table 5, *GS-LRM + 3D Warping* fails to handle new content and accumulates warping errors over time. A simple fusion baseline (*Concat + Dropout*) mitigates primitive growth by discarding 50% per frame but lacks refinement and often removes accurate primitives.

For the reported running time in Table 4, GS-LRM and ours are tested on a single H100 card. 3DGS, 3DGStream, and 4DGS are tested on a single A100 GPU due to limitations with the H100 card. While the efficiency of these optimization-based methods can be enhanced, they still do not achieve a frame rate of 1 FPS. Running the pre-trained 2D tracking model at keyframes every 5 frames increases the runtime from 0.07s to 0.35s. A detailed runtime breakdown of our method for both static and dynamic scenes is presented in Table 6.

## 6.3 Ablation Studies

**Ablation on Sparse Voxel Transformer.** We show model ablations on DL3DV in Table 2. Naive non-learning fusion of Gaussian attributes in each voxel significantly degrades results. The absence of a coarse-to-fine setup limits the capacity of the output Splats, and without sparsification, we downsampled the grid to 0.75x its original resolution to avoid exceeding memory capacity for the voxel transformer, both leading to poorer reconstruction quality. We also experiment with using a 3D CNN to downsample and upsample the voxel grid instead of our designed coarse-to-fine voxel hierarchy. This approach yields results approximately 1 dB worse than our final model and is significantly slower. We also find splat features helpful, operating on splat parameters directly leads to ∼1 dB drop in PSNR.

**Ablation on Temporal Components.** We conducted ablation studies on the Neural3DVideo dataset to evaluate our streaming reconstruction components, as shown in Table 7. Removing historical splats

Table 6: **Inference Runtime Breakdown of Our Method for Static and Dynamic Reconstructions.**

**Static Scenes** (256×256 Resolution)

|  | Multiview Transformer | Splat-to-Voxel Transfer | Voxel Transformer | Rendering |
|---|---|---|---|---|
| Time (s) | 0.026 | 0.006 | 0.015 | 0.003 |

**Dynamic Scenes** (320×240 Resolution)

|  | Multiview Transformer | 2D Tracking | 3D Warping | Fuse-and-Refine | Rendering |
|---|---|---|---|---|---|
| Time (s) | 0.030 | 0.028 | 0.018 | 0.021 | 0.003 |

Table 7: **Ablation on Temporal Components.** Incorporating historical information and our proposed mechanisms for leveraging it are crucial for achieving consistent and high-quality novel-view streaming results.

| Method | PSNR↑ | SSIM↑ | LPIPS↓ | Flicker↓ |
|---|---|---|---|---|
| w/o History | 27.17 | **0.890** | 0.107 | 5.91 |
| w/o 3D Warping | 27.12 | 0.884 | 0.107 | 6.09 |
| w/o Error-aware Fusion | 26.73 | 0.874 | 0.109 | 3.01 |
| Full Model | **27.41** | 0.886 | **0.104** | **2.92** |

Table 8: **Comparison Between Two-stage and Joint training.** Jointly training GS-LRM and our method achieves better performance than the sequential two-stage strategy that pre-trains GS-LRM.

| Method | PSNR↑ | SSIM↑ | LPIPS↓ |
|---|---|---|---|
| GS-LRM | 28.59 | 0.925 | 0.063 |
| Ours (Two-Stage) | 30.34 | 0.934 | 0.054 |
| Ours (Joint) | **30.61** | **0.935** | **0.052** |

for fusion failed to resolve temporal flicker. Disabling 3D warping caused abrupt keyframe changes and higher reconstruction errors. Removing error-aware fusion degraded quality due to warping artifacts and other errors.

**Ablation on Training Schemes.** Our framework supports both the two-stage training strategy described in Sec. 4.3 and joint training of the multi-view and voxel Transformers from scratch. As shown in Table 8, the joint training scheme can achieve better performance than the two-stage approach.

## 7 Conclusion

In summary, we introduce a novel Fuse-and-Refine module that can efficiently consolidate Gaussian primitives from multiple sources into a canonical and consistent representation. It enhances existing feed-forward models, and the design of the module naturally allows for reusing primitive information from past frames for better temporal coherence in streaming setups. Extensive evaluations show that our method achieves state-of-the-art performance on both static and dynamic datasets under sparse camera setups, delivering sharper reconstructions and reduced temporal flicker compared to previous feed-forward and optimization-based methods, all at interactive frame rates on a single GPU.

While the results of our approach are promising, it does have limitations. First, it depends on reasonably good initial primitives. For instance, in large-baseline setups such as the LongVolumetricVideo dataset, even the current best feed-forward model, GS-LRM, experiences a performance drop. As a result, our method yields limited improvements based on its input in such challenging scenarios. A potential solution is to train on a larger-scale dataset that covers a broader range of baseline configurations. Second, temporal artifacts still persist, particularly in regions with dynamic objects. The observed artifacts primarily arise from our current simple 3D warping strategy. Lifting 2D tracking into 3D from sparse views remains inherently error-prone and presents a persistent challenge in this ongoing field. A promising future direction is to investigate approaches that avoid explicit tracking or adopt adaptive online reconstruction strategies, potentially prioritizing regions with higher reconstruction errors.

**Acknowledgement.** We would like to express our gratitude to John Flynn, Kathryn Heal, Lynn Tsai, Srinivas Kaza, Qin Han, and Tooba Imtiaz for their helpful discussions, as well as Clément Godard, Jason Lawrence, Michael Broxton, Peter Hedman, Ricardo Martin-Brualla, and Ryan Overbeck for their valuable comments.

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

# A Implementation Details

Our framework is implemented in JAX [2] and trained on NVIDIA A100 GPUs. We use the Adam optimizer [24] with an initial learning rate of 4e-4, applying cosine learning rate decay with linear warmup. The warmup period is set to 5000 training steps.

## A.1 Network Structure

Our network consists of a *Multi-view Transformer* backbone and a *Sparse Voxel Transformer*. The multi-view transformer has 24 transformer layers with 1024 hidden dimensions and 16 attention heads. The sparse voxel transformer uses 6 layers with 128 hidden dimensions and 8 attention heads. To demonstrate the effectiveness of our representation, we use a 12-layer multi-view transformer in Table 4, showing that it can outperform GS-LRM by 2dB in PSNR while maintaining a comparable network capacity and similar inference time.

Each transformer layer [67] consists of a self-attention mechanism and feed-forward networks, with all bias terms removed following GS-LRM [84]. Additionally, following LVSM [21], we incorporate QK-Norm [19] to stabilize training. For efficiency, we utilize the CUDNN-FlashAttention [10] implementation available in Jax.

The shallow multi-layer perceptrons $\mathrm{MLP}_{coarse}$ and $\mathrm{MLP}_{fine}$ share similar structures with the feed-forward network in a transformer layer. It includes two linear layers with a LayerNorm [1] and GeLU [18] activation function, and an additional residual connection.

## A.2 Hybrid Splat-Voxel Representation

Our SplatVoxel model is inspired by Lagrangian-Eulerian representations in physical simulation where we combine the Gaussian splats (Lagrangian representation) and voxels (Eulerian representation) to assist warping and fusion of history information.

We use the official splatting-based rasterizer from 3D Gaussian Splatting [22] to render images from predicted Gaussian primitives. To ensure a fair comparison with GS-LRM on RealEstate10K[87], we set the spherical harmonics degree to 0 for Table 9. For models trained on the DL3DV dataset [35], the degree is set to 1 to enhance view-dependent effects.

Our voxel grid is aligned with the one target camera, and is in the normalized device space (NDC), where the z-axis is linear in disparity. This space is helpful in presenting unbounded, in-the-wild scenes. The full fine volume size is set to $[64, H, W]$, where the $H, W$ corresponds to the rendered image height and width. The coarse volume size is set to $[32, H//8, W//8]$, meaning each coarse-level volume is subdivided into $[2, 8, 8]$ fine-level volumes. The cost volume can also be constructed from any meaningful reference view, such as the average of the input cameras, to produce 3D Gaussian Splatting (3DGS) for free-viewpoint novel view synthesis. Moreover, the application of our sparse voxel transformer is not limited to cost volumes. It naturally generalizes to other voxel structures, such as octrees, making it applicable to broader scenarios like SLAM.

We initialize the voxels by depositing the Gaussian primitives to the volume with a 3D distance-based kernels used in Material Point Method [59]:

$$\mathcal{K}(\mathbf{x}) = K\left(\frac{x}{h}\right) K\left(\frac{y}{h}\right) K\left(\frac{z}{h}\right), \tag{10}$$

where $\mathbf{x} = (x, y, z)$ is the offset from the voxel center to the Gaussian primitive center, and $h$ is the grid size used to normalize the offset vector to the range $[-1, 1]$. The function $K(x)$ is a one-dimensional cubic kernel function given by:

$$K(x) = \frac{1}{6}(3|x|^3 - 6x^2 + 4) \tag{11}$$

Rather than directly generating new attributes, we design the voxel transformer to predict residuals that are added to the initial Gaussian rendering attributes on voxels. Specifically, given the input fine-level voxel attributes obtained through the Splat Fusion procedure $\mathbf{E}_j$ which comprise the fused

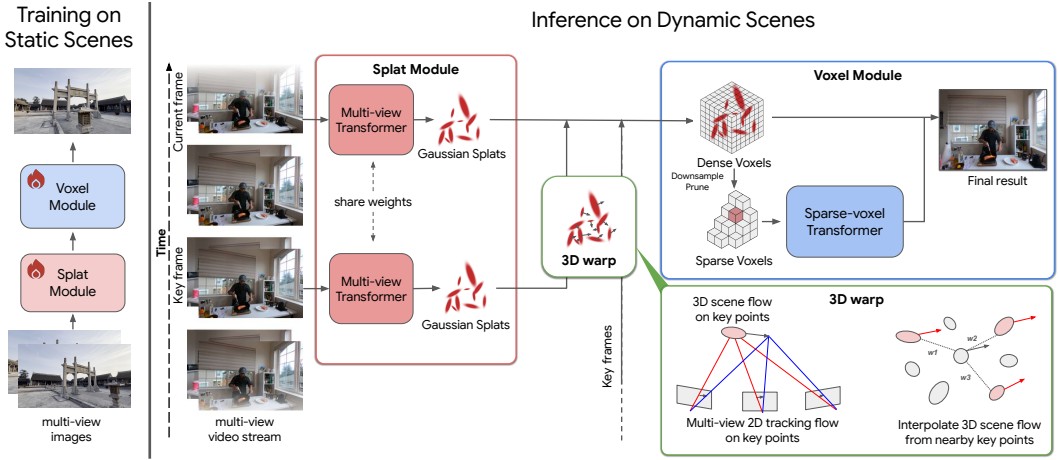

Figure 5: **History-aware Streaming System.** Our method can generalize to dynamic scenes during inference while trained only on static scenes. The hybrid Splat-Voxel model first extracts input image features using a multi-view transformer, which outputs pixel-aligned Gaussian splats for each input image with associated features. The splat features are then deposited onto a coarse-to-fine voxel grid using the decoded position, and a secondary sparse voxel transformer processes the grid features to output final Gaussian parameters. To merge history, we compute the triangulated scene flow from the input views and perform keypoint guided deformations. These deformed splats can be treated identically to the input-aligned splats, and be similarly deposited into voxel grid to merge the previous state with the current state.

splat feature $\mathbf{F}_j$ and rendering parameters $\mathbf{G}_j$ throught the Splat Fusion procedure, the transformer obtain the final splat rendering parameters $\mathbf{G}'_j$ in Eqn. 7 as:

$$\begin{aligned} \Delta\mathbf{G}_j &= \mathrm{MLP}_{fine}([\mathbf{O}_j, \mathbf{E}_j]) \\ \mathbf{G}'_j &= \mathbf{G}_j + \Delta\mathbf{G}_j. \end{aligned} \tag{12}$$

### A.3 History-aware Streaming Reconstruction

**Triangulation**    For solving the close closest point to the $N$ camera rays originating from the 2D projections $\mathbf{p}^i_t$, which can be formulated as the following least-squares optimization problem:

$$\mathbf{x}_c, \lambda_i = \arg\min_{\mathbf{x}, \lambda_i} \sum_i \|(\mathbf{o}_i + \lambda_i \mathbf{d}_i) - \mathbf{x}\|^2 \tag{13}$$

where $\mathbf{o}_i$ and $\mathbf{d}_i$ denote the origin and direction of the $i$-th camera ray, respectively, and $\lambda_i$ is a scalar representing the depth along the ray. This least-squares problems can be converted into solving an linear equation as

$$\left(\sum_i \gamma_i (I_3 - \mathbf{d}_i \mathbf{d}_i^T)\right)\mathbf{x} = \sum_i \gamma_i (I_3 - \mathbf{d}_i \mathbf{d}_i^T)\mathbf{o}_i \tag{14}$$

where $I_3$ denotes the three-dimensional identity matrix, and $\gamma_i$ is a visibility mask that indicates whether the projected 2D point is visible throughout the motion occuring in that view. The visibility mask is obtained through a combination of 2D tracking occlusion checks and a depth-based visibility verification, which involves comparing the rendered depth from Gaussian primitives with the projection depth to determine whether the point is visible in front of the surface.

We use TAPIR [11] as our 2D tracking backbone, which takes query points and video frames as input and produces 2D tracking positions throughout the video with the corresponding occlusion mask, indicating whether a tracked point is visible in a given frame. In addition to the occlusion mask obtained from 2D tracking, the visibility mask $\gamma_i$ used in Eqn. 14 for triangulation also depends on depth-based visibility. Depth-based visibility is determined by comparing the projected depth of a given splat with the depth map obtained by rendering all splats to a given view. If the projected depth

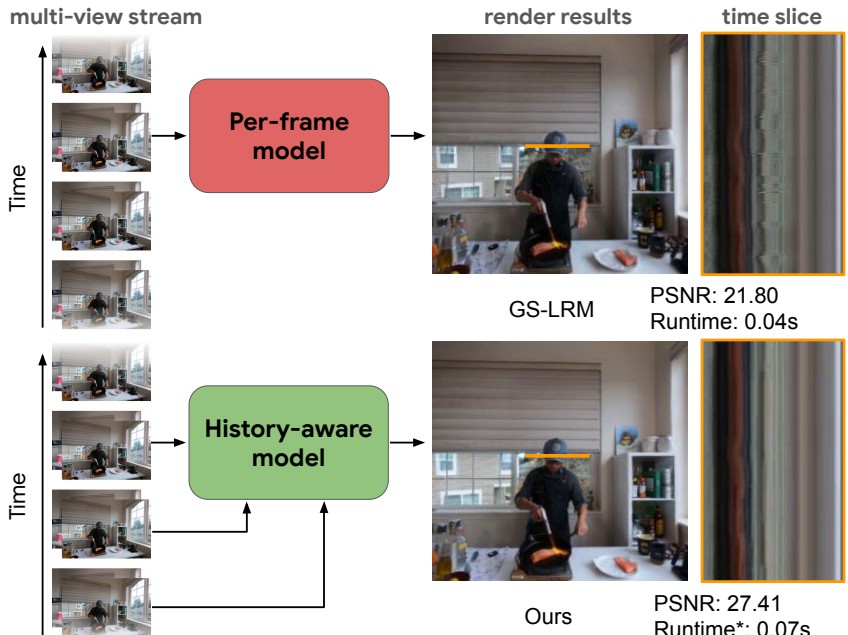

Figure 6: (Top) Per-frame reconstruction methods, which produce an independent scene reconstruction at each time step, are prone to flickering artifacts. (Bottom) In constrast, our history-aware novel view streaming model merges previous and current frame information, allowing us to better model occluded regions and improve temporal stability. Our method achieve state-of-the-art visual quality and temporal consistency, and runs in interactive rate (15 fps with a 350ms delay$^*$) on two view inputs of resolution $320 \times 240$.

is smaller, the splat is considered unoccluded. This process utilizes alpha blending of splat depths to derive rendered depths from 3D Gaussian Splatting, a widely adopted technique that has been demonstrated to be effective [8].

**Embedded Deformation Graph**    Assuming motion continuity between neighboring frames, we first estimate optical flow between each keyframe and its subsequent frame using RAFT [65] to identify regions of significant movement. We select regions within the input views where the optical flow norm exceeds a threshold of 0.2 pixels as areas of significant motion. Each splat is then projected onto the input views to determine whether its projected position falls within these significant motion regions and remains unoccluded based on the depth-based visibility verification aforementioned. If a splat exhibits significant motion in more than half of the views, it is prioritized for sampling as an anchor splat. During experiments, we found that uniformly sampling 512 points from the prioritized splats using the Farthest Point Sampling method [47] is sufficient for effectively handling moving Gaussians in the keyframes.

To warp splats from the previous frame, we first compute deformations on the anchor splats using triangulation, and then propagate the motion to all primitives through Linear Blend Skinning (LBS). For each Gaussian, we employ a K-nearest neighbors approach with $k = 4$ to identify the closest anchor splats. The position displacement $\delta \mathbf{p} \in \mathbb{R}^3$ is then determined by blending the deformations of the selected anchor points, denoted as $\mathbf{t}_i \in \mathbb{R}^3$.

$$\delta \mathbf{p} = \sum_{i=1}^{k} w_i \mathbf{t}_i$$
$$w_i = \frac{\exp(-\|d_i\|^2/\sigma^2)}{\sum_{j=1}^{k} \exp(-\|d_j\|^2/\sigma^2)},$$

(15)

where $d_i$ represents the Euclidean distance between the Gaussian primitive and the $i$-th anchor point, while $\sigma$ controls the smoothness of the weighting function, set to the mean distance between anchor points and the querying splats. To ensure that static splats remain unaffected, we set displacement

Table 9: **Reproduced GS-LRM Performance**. Our reproduced GS-LRM achieves improved LPIPS and SSIM scores with a slight decrease in PSNR on the RealEstate10K dataset compared to the officially reported numbers. We also include comparisons with other feed-forward novel view synthesis methods in this table.

| Method | PSNR ↑ | SSIM ↑ | LPIPS ↓ |
|---|---|---|---|
| pixelNeRF [82] | 20.43 | 0.589 | 0.550 |
| GPNR [60] | 24.11 | 0.793 | 0.255 |
| Du et al. [13] | 24.78 | 0.820 | 0.213 |
| GS-LRM (Official) [84] | 28.10 | 0.892 | 0.114 |
| GS-LRM (Reproduced) | 27.78 | 0.902 | 0.082 |
| GS-LRM (Reproduced) + **Ours** | **28.47** | **0.907** | **0.078** |

Table 10: **Inference Time on RealEstate10K.** We compare the inference time of our method with other feed-forward Gaussian Splatting approaches. All runtimes are measured on an NVIDIA A100 GPU. The slight runtime overhead of our method relative to GS-LRM can be mitigated by reducing the number of multi-view transformer layers. As reported in Table 2, our method improves PSNR by over 2 dB while maintaining comparable model size and inference time.

| Method | PSNR↑ | SSIM↑ | LPIPS↓ | Inference Time(s) |
|---|---|---|---|---|
| pixelSplat [4] | 26.09 | 0.863 | 0.136 | 0.104 |
| MVSplat [6] | 26.39 | 0.869 | 0.128 | 0.044 |
| GS-LRM [84] | 28.10 | 0.892 | 0.114 | 0.041 |
| Ours | **28.47** | **0.907** | **0.078** | 0.067 |

$\delta \mathbf{p}$ to be zero if the distance to the anchors is greater than $\lambda \bar{d}$, where $\bar{d}$ is the mean distance between anchor points and $\lambda$ is a hyperparameter controlling the motion graph's extent. In practice we set $\lambda$ between 0.2 to 0.5 depending on the scene.

For efficiency, we only maintain primitives from a set of past keyframes, and the contribution of these primitives are smoothly adjusted as older keyframes are replaced by newer ones to ensure seamless temporal transitions. Due to time constraints from querying the 2D tracking model, in practice we use two keyframes placed 5 frames apart, each lasting for 10 frames.

**Error-aware Streaming Fusion**   To further mitigate artifacts resulting from warping inaccuracies of historical primitives (e.g. warping error, appearance variation, or new objects), we use an adaptive fusion weighting mechanism during splat fusion based on the per-pixel error between the past and current frame splats when rendered to the input views. In more detail, each historical splat is projected onto the input views to assess whether it produces a higher rendering error and appears in front of the rendered depth of the current frame splats. If a historical splat results in increased error in more than half of the input views, we set its fusion weight to zero to suppress unreliable contributions. Concurrently, current frame splats predicted from the multi-view transformer that increase rendering error are assigned zero weight. To enhance temporal continuity, we retain 50% of the voxel attributes from the past keyframe in static areas. Static areas are identified during the Splat-to-voxel transfer process, where historical splats undergoing deformation mark their neighboring voxels as regions of motion.

Fig. 5 shows the system implementation of our history-aware streaming reconstruction. The streaming fusion and refinement process enables our model to significantly reduce temporal flickering and improve reconstruction quality, as demonstrated in Fig. 6.

# B   Dataset

## B.1   Static Scene Datasets

**RealEstate10K [87]**   For the RealEstate10K dataset, we set the training and testing resolution to $256 \times 256$. Both GS-LRM and our model are trained for 100K iterations. Specifically, we first train

Table 11: **Quantitative Comparison with 4DGS and GS-LRM.** Evaluation across all sequences in LongVolumetricVideo [79] and Neural3DVideo [27]. The best and second-best results are highlighted.

| Dataset | Sequence | 4DGS | | | | GS-LRM | | | | Ours | | | |
|---|---|---|---|---|---|---|---|---|---|---|---|---|---|
| | | PSNR ↑ | SSIM ↑ | LPIPS ↓ | Flicker ↓ | PSNR ↑ | SSIM ↑ | LPIPS ↓ | Flicker ↓ | PSNR ↑ | SSIM ↑ | LPIPS ↓ | Flicker ↓ |
| LongVolCap | *Corgi* | 17.25 | 0.530 | 0.423 | 8.17 | 26.24 | 0.882 | 0.118 | 9.95 | 26.48 | 0.885 | 0.109 | 8.76 |
| | *Bike* | 20.26 | 0.742 | 0.288 | 4.89 | 24.74 | 0.830 | 0.158 | 6.97 | 24.64 | 0.844 | 0.140 | 6.80 |
| Neural3DV | *coffee_martini* | 19.29 | 0.676 | 0.268 | 1.83 | 20.86 | 0.801 | 0.149 | 7.21 | 23.10 | 0.832 | 0.126 | 2.33 |
| | *cut_roasted_beef* | 25.41 | 0.851 | 0.167 | 3.17 | 21.95 | 0.855 | 0.134 | 5.51 | 29.15 | 0.909 | 0.096 | 3.26 |
| | *flame_steak* | 26.54 | 0.854 | 0.159 | 3.24 | 22.04 | 0.866 | 0.121 | 5.56 | 29.10 | 0.903 | 0.100 | 3.31 |
| | *cook_spinach* | 24.41 | 0.794 | 0.198 | 3.49 | 21.93 | 0.872 | 0.119 | 5.78 | 29.26 | 0.906 | 0.103 | 3.42 |
| | *flame_salmon_1* | 19.92 | 0.709 | 0.250 | 2.18 | 21.70 | 0.827 | 0.126 | 5.65 | 24.70 | 0.860 | 0.102 | 2.52 |
| | *sear_steak* | 23.42 | 0.804 | 0.206 | 2.52 | 22.23 | 0.869 | 0.122 | 4.92 | 29.14 | 0.907 | 0.098 | 2.64 |

Table 12: **Method Comparison.** Unlike existing baselines, our framework uniquely supports the three key requirements of the novel view streaming task targeted in this paper: fast reconstruction with interactive rate , sparse view inputs, and history awareness. In particular, our streaming approach only uses *past* frames, in contrast to DeformableGS [81] and DeformableNerf [44] that optimize over *all*[*] frames. The history-awareness in our model mitigates temporal flickering and occlusion artifacts, and is also able to support unlimited sequence lengths.

| Method | Interactive Speed | Sparse-view Input | History Awareness |
|---|---|---|---|
| NeRF [41] | ✗ | ✗ | ✗ |
| InstantNGP [42] | ✗ | ✗ | ✗ |
| 3DGS [22] | ✗ | ✗ | ✗ |
| DeformableGS [81] | ✗ | ✓ | ✓[*] |
| DeformableNerf [44] | ✗ | ✓ | ✓[*] |
| PixelSplat [4] | ✓ | ✓ | ✗ |
| GS-LRM [84] | ✓ | ✓ | ✗ |
| Quark [15] | ✓ | ✓ | ✗ |
| **Ours** | ✓ | ✓ | ✓ |

our multi-view transformer backbone for 80K iterations, followed by 20K iterations of joint training with the voxel decoder. Each training batch consists of two input views and six target views with a baseline of one unit length between the input views, following the training setup of GS-LRM. For evaluation, we use the same input and target indices as PixelSplat and GS-LRM.

**DL3DV [35]** For the DL3DV dataset, we set the training and testing resolution to $384 \times 216$. For training, we randomly select one image in the scene as the target, and randomly select four of the nearest eight cameras to the target as inputs. We scale the scene such that the cameras fit within a unit cube. For evaluation, we use every eighth image as the target set, and for each target we use the nearest four cameras not in the target set as inputs. We average metrics per scene, and then average over all scenes. In our experiments on the DL3DV test set (Table 2), both our model and GS-LRM are trained using 8 GPUs, with a batch size of 2 per GPU, for a total of 200K iterations. Our method supports both joint training of the multi-view Transformer and voxel decoder from scratch, and a two-stage training scheme. As shown in Table 8, joint training from scratch outperforms the two-stage approach, where the multi-view Transformer is first trained for 100K iterations, followed by another 100K iterations of joint training with the voxel decoder. Figure 7 illustrates that after training the multi-view Transformer used in GS-LRM, our proposed sparse voxel Transformer converges to significantly improved results with only a small number of additional training steps.

For cross-dataset evaluation on dynamic scene datasets, we train both GS-LRM and our model on DL3DV using a batch size of 128, for a total of 300K iterations. The training follows a two-stage process: we first train the multi-view transformer backbone for 200K iterations, then jointly train the voxel decoder for 100K iterations.

Table 13: **Comparison with Gaussian Graph Network (GGN) [86]**. Both models are trained on RealEstate10K (2-view training, 4-view testing).

| Method | PSNR↑ | SSIM↑ | LPIPS↓ |
|---|---|---|---|
| GGN [86] | 24.76 | 0.784 | 0.172 |
| **Ours** | **28.79** | **0.914** | **0.081** |

Table 14: **Extension to DepthSplat.** Our method can be seamlessly integrated into other feed-forward 3D Gaussian Splatting frameworks to enhance their performance. Both models are trained on a small subset of DL3DV (2-view training, 4-view testing).

| Method | PSNR↑ | SSIM↑ | LPIPS↓ |
|---|---|---|---|
| DepthSplat [76] | 16.57 | 0.4108 | 0.4272 |
| DepthSplat + **Ours** | **17.66** | **0.4426** | **0.4019** |

Figure 7: **Comparison of Multi-View and Voxel Transformers.** Validation curves on the DL3DV dataset show that our 3D Sparse Voxel Transformer converges faster and achieves significantly better final performance when initialized with a pre-trained 2D Multi-View Transformer, compared to training with the 2D Multi-View Transformer alone. Our method also supports joint training of the Multi-View and Voxel Transformers, leading to further performance improvements as shown in Table 8.

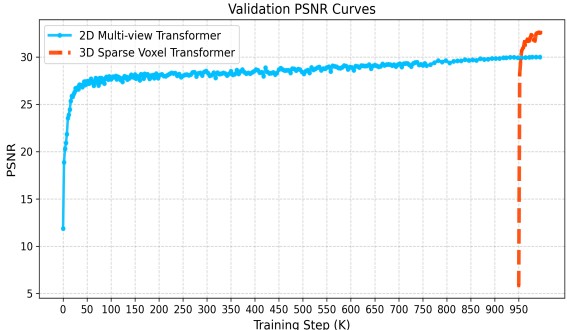

## B.2 Dynamic Scene Datasets

For evaluation, we use two input views with a small-baseline setup and across six sequences from Neural3DVideo, and four input views with a large-baseline setup and 1,500 frames for each of the two sequences from LongVolumetricVideo.

**Neural3DVideo [27]** We use six sequences from the dataset for evaluation, each containing 290 frames. The center target camera is selected as the evaluation view, while the two second-closest cameras on either side are used as input cameras. We use resolution $320 \times 240$ downsampling from their original resolution.

**LongVolumetricVideo [79]** For evaluation, we use the Corgi and Bike sequences and select a 4-input-camera setup that covers one target view. We failed to find enough baseline coverage for other sequences in LongVolCap using only 4 input cameras. For both the Corgi and Bike sequences, we use 1500 frames for evaluation. We use resolution $384 \times 216$ for Corgi and $256 \times 256$ for the Bike sequences, downsampling from their original resolution.

## C Baselines

**3DGS** We use the original 3D Gaussian Splatting implementation [22], and reduce the number of training iterations from original 30,000 to 1,000 for *Neural3DVideo* and 1,500 for *LongVolumetricVideo*. This modification not only accelerates training but also significantly enhances novel view synthesis results (e.g. around 4 PSNR improvement). Given the sparsity of the input data, this adjustment is crucial for preventing overfitting. All other parameters are kept consistent with the original implementation.

**3DGStream** We use the open-source 3DGStream [63] implementation to process the multi-view videos. We first use the original 3DGS to optimize on the first frame for 1,500 iterations, warm up the NTC cache, and then optimize on the next frame using the previous 3DGS for 150 iterations, following the original implementation.

**4DGS** We use the open-source 4DGS [74] implementation to process the multi-view videos. Following their implementation, we first optimize the first frame for 3,000 iterations, and then optimize on the full multi-view video sequence for 14,000 iterations.

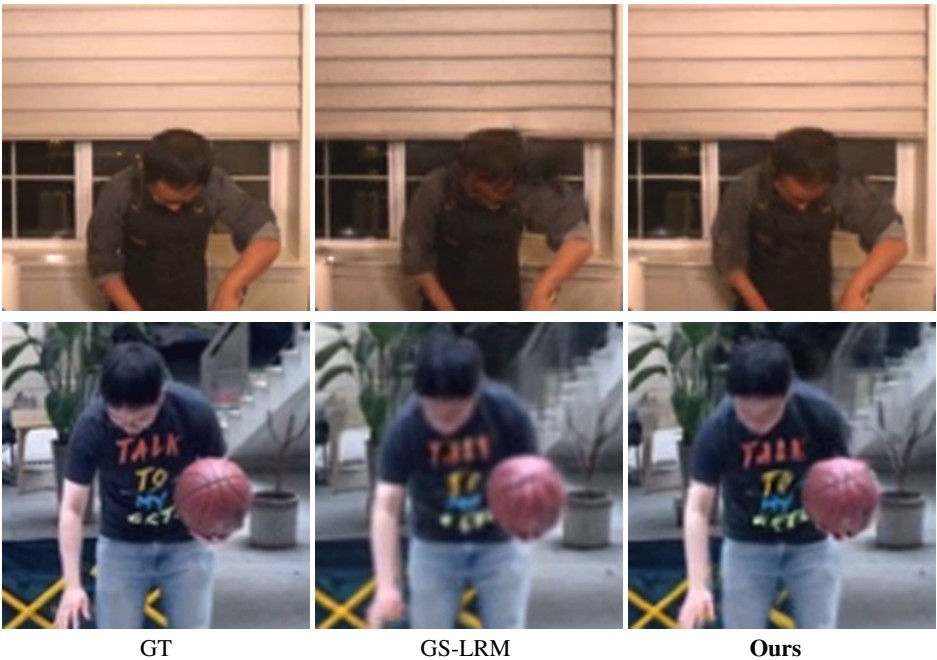

| GT | GS-LRM | **Ours** |

Figure 8: **Reconstruction Close-ups on Dynamic Scenes.** We show zoomed-in novel view reconstructions of GS-LRM and our model on dynamic scene datasets. Our model better handles occlusion boundaries with sharper detail.

**GS-LRM** We enhance GS-LRM's speed by replacing the CNN deconvolution with an MLP unpatchify layer as proposed in [21], and implementing it in JAX. We maintain the same network parameters and training loss weights as in the original paper. Additionally, we use the LPIPS loss employed in our model, achieving improved LPIPS and SSIM scores, with a slight decrease in PSNR, as detailed in Table 10. We adopt the same color modeling approach as the official paper for RealEstate10K, setting the spherical harmonics degree to 0. For the GS-LRM baseline trained on DL3DV, we set the degree to 1 to ensure a fair comparison with our method. Additionally, we found that this modification enhances GS-LRM's performance on DL3DV.

# D   Additional Results

**Novel View Streaming** Table 11 presents detailed metrics for our results alongside the two best-performing baselines, 4DGS and GS-LRM, evaluated across all eight sequences. We show close-ups of the reconstruction in Fig. 8 comparing ours with GS-LRM.

**Feed-forward Novel View Synthesis** We also provide additional examples comparing GS-LRM with our method on both DL3DV and RealEstate10K, as shown in Fig. 9 and Fig. 10.

We also compare our method with Gaussian Graph Network (GGN) [86] on the RealEstate10K dataset, following its experimental configuration. Both models are trained with 2 input views and evaluated with 4 input views to assess generalization across varying input configurations. As shown in Table 13, our method significantly outperforms this state-of-the-art Gaussian fusion approach.

We further validate the generalization capability of the proposed Fuse-and-Refine module by integrating it into DepthSplat [76]. We retrain the publicly available DepthSplat implementation on the DL3DV dataset at a reduced resolution of $128 \times 192$. The training subset includes 1,000 scenes with 2 input views, and evaluation is performed on 10 test scenes with 4 input views. Table 14 shows that incorporating our module leads to a performance gain over the original DepthSplat, demonstrating the adaptability of our approach.

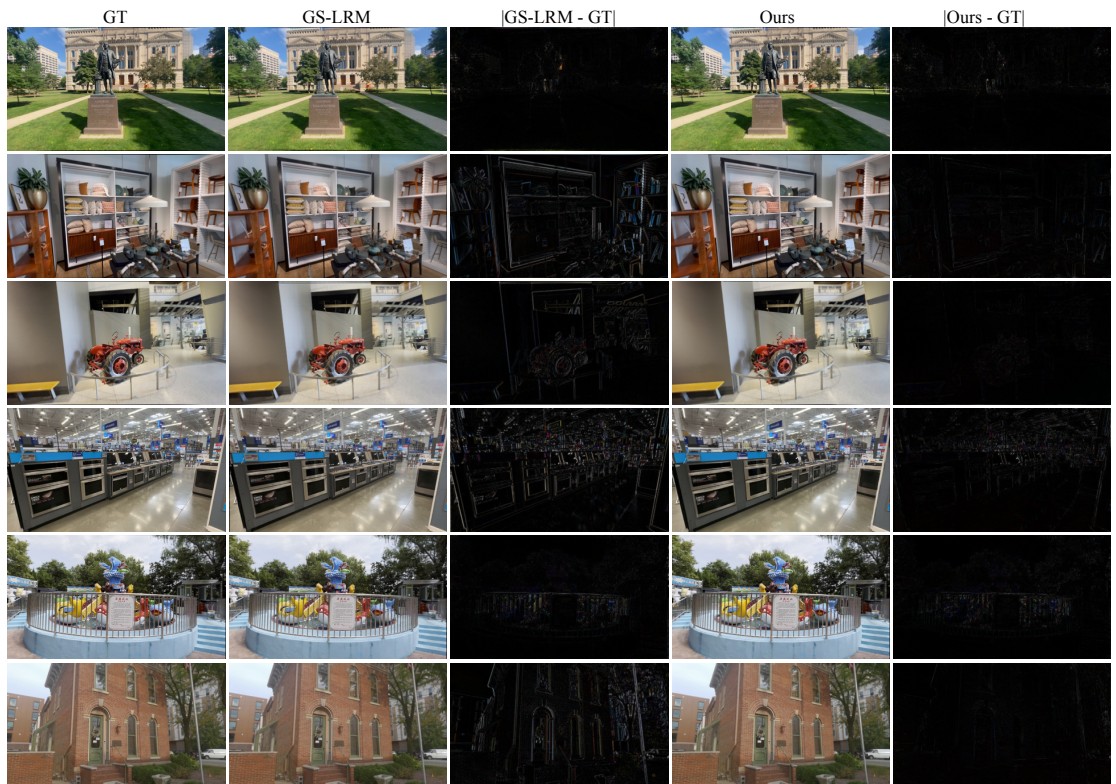

Figure 9: **DL3DV Static Scene Reconstruction.** Additional qualitative examples of GS-LRM and our method trained on DL3DV.

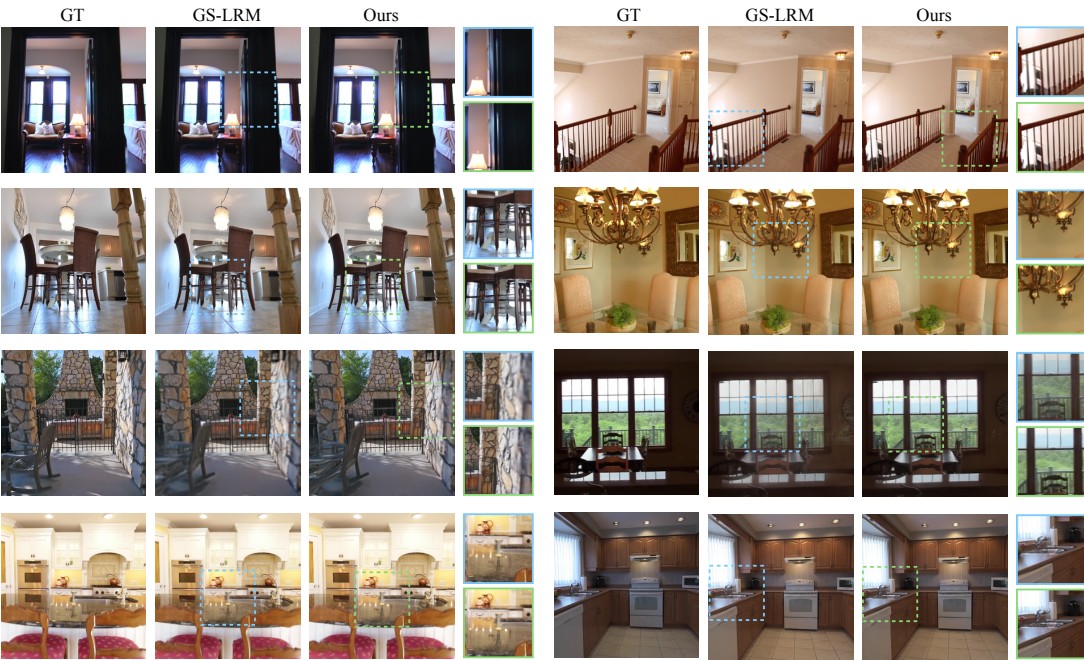

Figure 10: **Re10K Static scene reconstruction.** Additional qualitative examples of GS-LRM trained on RealEstate10k [87].

# E    Societal Impact

In this section, we reflect on the broader societal implications of our work. Our method facilitates fast and high-quality 3D reconstruction from sparse views, significantly lowering the barrier for creating digital assets from real-world scenes or objects. This has potential benefits across domains such as digital content creation, robotics, autonomous navigation, cultural heritage preservation, and immersive applications in AR/VR. Additionally, it supports real-time streaming use cases, toward fully immersive experiences.

However, like many 3D/4D reconstruction methods, our approach could be misused to create detailed models of people, places, or objects without consent, raising concerns about privacy, identity theft, and misleading content. While our work is intended for positive and legitimate use, we acknowledge the ethical responsibilities involved and urge users to apply this technique responsibly.

