# OpenReview forum: "Learning Efficient Fuse-and-Refine for Feed-Forward 3D Gaussian Splatting"
_NeurIPS.cc/2025/Conference — NeurIPS 2025 poster_

### Official Review · Reviewer_Wi3w · 2025-06-16

**Clarity:** 3
**Significance:** 3
**Originality:** 3
**Rating:** 3
**Confidence:** 4

**Summary:**

This paper proposes a Fuse-and-Refine module to improve feed-forward 3D Gaussian Splatting for both static and dynamic scene reconstruction. The method introduces a hybrid Splat-Voxel representation that aggregates and refines Gaussian primitives using a sparse voxel transformer. It enables efficient fusion of overlapping or temporally adjacent primitives, significantly improving reconstruction quality and temporal consistency. Notably, the model generalizes to dynamic scenes in a zero-shot manner, achieving interactive performance without retraining.

**Questions:**

1.  Could the authors elaborate on how their method fundamentally differs from SplatFormer? Specifically, what architectural or functional advantages does the proposed Fuse-and-Refine pipeline offer over that prior work?

2. Could you provide further analysis or experimental evidence to explain why the performance improvement is more pronounced in dynamic scenes than in static ones? For instance, does the temporal fusion mechanism contribute more significantly than the spatial refinement in achieving this gain?

3. Could you provide a detailed breakdown of the computational bottlenecks introduced by your method? Have you explored any optimization strategies to reduce runtime while preserving the benefits, especially in dynamic scene reconstruction?

4. Could you include experiments or analysis showing how your proposed modules perform when integrated with other state-of-the-art 3D Gaussian Splatting frameworks beyond GS-LRM? This would help demonstrate the generalizability and flexibility of your approach.

**Ethical Concerns:**

["NO or VERY MINOR ethics concerns only"]

**Final Justification:**

My primary concerns regarding the “Comparison with SplatFormer” and “Integration with other Frameworks” sections remain unaddressed.

1. Naming Convention of Point Transformer: The term “Point Transformer” conventionally refers to the seminal work by Zhao et al. [1], which introduced the foundational framework for point cloud processing via transformer architectures. In contrast, SplatFormer [2] represents a distinct, recently proposed extension tailored for 3D Gaussian splatting. This distinction should be explicitly clarified to avoid terminological ambiguity and ensure proper attribution.

2. Hierarchical Aggregation in Point-based Designs: Point Transformer [1] have demonstrated hierarchical feature learning through iterative k-nearest neighbor (k-NN) aggregation and multi-scale sampling strategies. A more rigorous discussion contrasting the proposed hierarchical mechanism with the proposed approach would strengthen the technical contribution.

3. Limited Validation of Framework Integration: While the integration with GS-LRM [3] is presented, the evaluation lacks generality to substantiate the method’s broader compatibility. To better demonstrate extensibility, experiments integrating with established frameworks should be included. This would more convincingly validate the architectural flexibility and scalability of the proposed approach.

[1] Zhao H, Jiang L, Jia J, et al. Point transformer[C]//Proceedings of the IEEE/CVF international conference on computer vision. 2021: 16259-16268.

[2] Chen Y, Mihajlovic M, Chen X, et al. Splatformer: Point transformer for robust 3d gaussian splatting[J]. arXiv preprint arXiv:2411.0

[3] Zhang K, Bi S, Tan H, et al. Gs-lrm: Large reconstruction model for 3d gaussian splatting[C]//European Conference on Computer Vision. Cham: Springer Nature Switzerland, 2024: 1-19.

Overall, I tend to keep my rating (Borderline reject).

**Limitations:**

Yes. The authors have clearly acknowledged both technical limitations, such as reliance on high-quality initial primitives and residual temporal artifacts, and potential societal risks, including privacy concerns from unauthorized 3D reconstruction. They also propose future directions and ethical considerations, demonstrating responsible awareness. Further improvement could include outlining safeguards to mitigate misuse.

**Paper Formatting Concerns:**

Please align Table 1 and Figure 2.

**Quality:**

3

**Strengths And Weaknesses:**

**Strengths**

1. The paper is clearly written and well structured.
2. The hybrid Splat-Voxel representation, coupled with the sparse voxel transformer, efficiently refines \~20 k Gaussian primitives in \~15 ms and sustains interactive rendering (\~15 FPS) while preserving detail, striking a strong balance between fidelity and computational cost.
3. The zero-shot history-aware streaming fusion generalizes to dynamic scenes without retraining and yields state-of-the-art PSNR, SSIM, and LPIPS on both static and dynamic benchmarks, highlighting the method’s practical impact.

**Weaknesses**

1. The paper claims novelty in introducing a Splat-to-Voxel transfer followed by Transformer-based refinement, but the proposed design closely resembles SplatFormer \[1], which is not cited or discussed. This omission raises concerns about the originality of the approach and the authors’ positioning of their contributions within the current literature.

2. The method demonstrates more significant performance gains on dynamic scenes than on static ones, yet the explanation for this difference is shallow. The paper lacks in-depth analysis or ablation to clarify how design components like temporal fusion or the hybrid representation contribute to dynamic reconstruction advantages, and why static scenes benefit less.

3. While the integration with GS-LRM shows promising improvements, it incurs substantial computational overhead, which undermines claims of real-time performance. The paper does not provide a breakdown of the time-consuming components or suggest optimizations to reduce latency and memory usage, especially important for practical deployment.

4. The method’s evaluation is limited to GS-LRM and GS-LAM, making it difficult to assess generalizability. There is no validation on other recent 3D Gaussian Splatting frameworks, which weakens the claim that the proposed modules are broadly applicable across different architectures.

5. The visual results do not adequately demonstrate the model’s handling of challenging cases such as occlusion or temporal inconsistency. More targeted visualizations and metrics are needed to evaluate robustness in synthesizing coherent views under occlusion or motion blur.

6. Several technical terms and components lack sufficient clarity. For example, “history-aware streaming reconstruction” is not properly defined; the semantic role of the feature vector $F_k$ is unspecified; and the voxel pruning strategy (retaining top 20% by weight) is a fixed heuristic without adaptive guidance or performance feedback.

7. The learning-based fusion strategy lacks theoretical grounding or interpretability analysis. Similarly, the error-aware fusion applies fixed thresholds to discard noisy primitives, but this may also remove useful information, potentially limiting reconstruction quality in complex or noisy scenes.

8. The paper does not report inference-time GPU memory usage or system footprint, which is critical for assessing feasibility on mobile or edge devices. Moreover, minor presentation issues—such as incorrect references in Table 1 and missing discussion of related works like Mini-Splatting [2], Scaffold-GS [3], and Octree-GS [4], should be corrected to improve rigor and completeness.

[1] Chen Y, Mihajlovic M, Chen X, et al. SplatFormer: Point Transformer for Robust 3D Gaussian Splatting[J]. arXiv preprint arXiv:2411.06390, 2024.

[2] Fang G, Wang B. Mini-splatting: Representing scenes with a constrained number of gaussians[C]//European Conference on Computer Vision. Cham: Springer Nature Switzerland, 2024: 165-181.

[3] Lu T, Yu M, Xu L, et al. Scaffold-gs: Structured 3d gaussians for view-adaptive rendering[C]//Proceedings of the IEEE/CVF Conference on Computer Vision and Pattern Recognition. 2024: 20654-20664.

[4] Ren K, Jiang L, Lu T, et al. Octree-gs: Towards consistent real-time rendering with lod-structured 3d gaussians[J]. arXiv preprint arXiv:2403.17898, 2024.

---

> ### Author Rebuttal · Authors · 2025-07-31
>
> We appreciate the reviewer’s valuable feedback and thoughtful questions. We will correct the formatting issues and incorporate the relevant related work in the revised version. Below, we address the discussion points in detail.
>
>
> ## **W1&Q1: Comparison with SplatFormer**
> We respectfully note that SplatFormer is discussed and cited in line 201. However, we are happy to further elaborate here and will expand the discussion in the related work section of the revised version. Notably, our design yields two significant benefits:
>
> **Fusion capability and applicability to dynamic scenes:** Our fuse-and-refine sparse voxel transformer is capable of fusing and refining an arbitrary number of input primitives into a fixed number of output tokens. In contrast, SplatFormer uses a Point Transformer that outputs the same number of Gaussian primitives as its input, without fusing nearby points spatially. As a result, SplatFormer is not suitable for online streaming reconstruction tasks, as it cannot fuse redundant primitives over time. Our hybrid splat-voxel representation supports efficient and flexible fusion in both space and time, a capability that SplatFormer's purely point-based design cannot support.
>
> **Effectively Handling of unbounded scene:** SplatFormer’s Point Transformer is sensitive to the spatial distribution of input points and is primarily designed for object-centric settings. According to its original paper, SplatFormer demonstrates limited scalability when applied to large, unbounded scenes. Its uniform downsampling strategy in the point Transformer disproportionately reduces the resolution of foreground objects relative to expansive backgrounds, making it challenging to preserve fine-grained details in complex environments. In contrast, our method employs a coarse-to-fine voxel hierarchy, with the depth dimension of the voxels representing equal increments in disparity, and  prunes empty space where appropriate to reallocate compute to occupied regions. Our design enables effective and scalable reconstruction of unbounded scenes, while achieving significantly higher efficiency. When processing a similar number of primitives, our method completes a forward pass in under 15 ms, compared to over 100 ms for SplatFormer.
>
>
>
>
>
>
>
> ## **W2&Q2:  Better Improvement on Dynamic Scenes**
> When applied to dynamic scenes, our fusion-and-refine module operates in both spatial and temporal domains. In addition to the multi-view spatial fusion-and-refinement also used in static scenes, it leverages temporal information by incorporating historical reconstructions.
> That said, the more pronounced improvements observed in dynamic scenes can be attributed to the combined effects of both spatial and temporal fusion, as demonstrated in the ablation study presented in Table 3. Specifically, adding the hybrid representation for spatial fusion and refinement without temporal information (Ours without History-aware Fusion) achieves a reconstruction PSNR of 27.17, compared to GS-LRM’s 21.80. This indicates that spatial fusion is the primary contributor to the overall improvement in reconstruction quality. However, this variant suffers from severe flickering artifacts due to the lack of temporal consistency, with a flicker score of 5.91 worse than GS-LRM’s 5.71. By incorporating temporal fusion, our full method further improves the reconstruction quality from 27.17 to 27.41 PSNR and significantly reduces flicker from 5.91 to 2.92.  These results underscore the essential role of temporal fusion in maintaining temporal consistency, which is critical for high-quality dynamic scene reconstruction. Overall, we emphasize that the effectiveness of our fusion-and-refinement module lies in its ability to operate jointly across spatial and temporal domains, which is a key strength of our approach.
>
>
> ## **W3&Q3: Efficiency Analysis**
>
>
> Here, we present a detailed breakdown of the inference runtime of our method for both static scene and streaming reconstruction, evaluated on an NVIDIA H100 GPU. We begin by reporting the runtime of individual components for static scene reconstruction on the RealEstate10K dataset, using two input views at a resolution of 256×256.
>
> **Table: Runtime Breakdown of Static Scene Reconstruction on the RealEstate10K Dataset**
>
> |Components|Multi-view Transformer|Splat-to-Voxel Transfer|Voxel Transformer|Rendering|
> |:-:|:-:|:-:|:-:|:-:|
> |Time(s)|0.026|0.005|0.016|0.003|
>
>
> We further analyze the runtime of each component in a streaming reconstruction setting on the Neural3DVideo dataset, using 2 input views at a resolution of 320×240. Note that 2D tracking is applied only to keyframes, which are sampled every 5 frames.
>
> **Table: Runtime Breakdown of Streaming Reconstruction on the Neural3DVideo Dataset**
>
> |Components|Multi-view Transformer|2D Tracking|3D Warping|Temporal Fuse-and-Refine|Rendering|
> |:-:|:-:|:-:|:-:|:-:|:-:|
> |Time(s)|0.030|0.28|0.011|0.026|0.003|
>
>
> The slight runtime overhead of our method compared to GS-LRM for static scene reconstruction can be mitigated by reducing the number of multi-view transformer layers in the GS-LRM backbone. As shown in Table 1 of our paper, our method **improves PSNR by over 2 dB while maintaining comparable model size and inference time**.
>
> For streaming reconstruction, the primary bottlenecks are the 2D tracking and 3D warping modules. We believe that adopting a faster 2D tracking backbone and exploring more advanced 3D warping strategies beyond our current simple warping algorithm represent promising directions for future work.
>
>
> ## **W4&Q4: Integration with Other frameworks**
> Our method can be naturally integrated with other feed-forward 3D Gaussian Splatting frameworks. Since the fuse-and-refine module directly operates on 3D primitives, it can be seamlessly plugged into the training pipelines of various feed-forward methods. In this paper, we choose GS-LRM as the backbone due to its state-of-the-art performance among existing feed-forward methods (see Table 2 of our paper).
> Due to time constraints, we were unable to include experiments with alternative backbones in the rebuttal. However, we plan to explore such integrations to further demonstrate the generalizability of our approach.
>
> ## **W5: Occlusion Handling and Temporal Consistency**
> We evaluate temporal consistency using the flicker metric, as described in Line 270 of the paper. As shown in Table 4, our method achieves the highest reconstruction quality while maintaining strong temporal consistency. A key challenge in evaluating occluded regions is obtaining accurate occlusion masks. We would be happy to include this metric and would appreciate any suggestions for generating reliable masks in sparse, in-the-wild videos.
>
> As illustrated in Figure 4, GS-LRM struggles to reconstruct the window due to occlusion by the person. In contrast, our method achieves consistent reconstruction by leveraging historical information to effectively handle occlusions, as also shown at 2:00 in the supplementary video.
>
>
> ## **W6: Clarification of Technical Terms**
> We thank the reviewer for pointing out these unclear terms and will revise them accordingly. The term history-aware streaming reconstruction refers to leveraging information from past frames to enhance the reconstruction of new frames in arbitrarily sparse-view, long streaming inputs, thereby improving both quality and temporal stability.
> The feature vector $F_k$ is introduced in Line 135 as additional splat features derived from the image to enhance the fuse-and-refine module. Its effect is analyzed in paper Table 1 under the ablation setting "Ours (w/o Splat Feature)."
>
> ## **W7: Interpretability Analysis of the Fuse-and-refine Strategy**
>
> The motivation behind our learning-based fusion strategy is to merge redundant primitives across both spatial views and temporal frames. As discussed in Line 140, deriving an analytical solution for fusing Gaussian primitives is highly challenging. Our fuse-and-refine approach addresses this by learning a global fusion mechanism from large-scale data using a voxel Transformer backbone. Given a set of initial Gaussian primitives, we aggregate local features from Gaussian primitives into a voxel representation, and then apply the Transformer to generate refined Gaussian primitives from voxel features. Through end-to-end training with photometric rendering supervision, the module is optimized to refine imperfect primitives into a set of high-quality Gaussians. We empirically validate the effectiveness of our fusion strategy through comparisons with GS-LRM in both static and streaming scene reconstruction, and present a comprehensive ablation study of its design components in paper Table 1.
>
> Our voxel sparsification strategy is motivated by the observation that large-scale unbounded scenes are typically sparse. The sparsification ratio (20% threshold) is chosen to keep the number of tokens below 10K, ensuring that the transformer runs within 15 milliseconds. In practice, we find that a sparsification ratio between 15% and 25% offers a good balance between performance and quality.  A promising direction for future work is to develop a more adaptive, scene-aware pruning strategy.
>
> The error-aware fusion module is designed to filter out primitives with large warping errors by comparing the rendered results of warped primitives against ground truth. We demonstrate the efficacy of this module in paper Table 3. Since useful primitives generally produce low errors, they are more likely to be preserved. Additionally, trained on large-scale datasets with diverse types of Gaussian primitive inputs, our learned fuse-and-refine strategy demonstrates robustness to variations in input primitives.
>
> ## **W8: Runtime GPU Memory**
>
> Our method consumes approximately 10 GB of GPU memory during inference, primarily due to the naive implementation of voxel grid storage. This can potentially be optimized by adopting a sparse data structure.

---

> > ### Author Response · Authors · 2025-08-05
> > **Official Comment by Authors**
> >
> > Dear Reviewer Wi3w,
> >
> > We would like to sincerely thank you for your valuable feedback and the time you’ve dedicated to reviewing our paper.
> >
> > As the discussion phase approaches its end, we hope our responses have addressed your concerns. If there are any remaining issues or questions, we would be happy to clarify further. If you find our revisions satisfactory, we would be truly grateful if you would consider raising your score for our paper.
> >
> > Thank you once again for your thoughtful comments and support throughout the review process.

---

> > ### Comment · Reviewer_Wi3w · 2025-08-06
> >
> > My primary concerns regarding the “Comparison with SplatFormer” and “Integration with other Frameworks” sections remain unaddressed.
> >
> > 1. Naming Convention of Point Transformer: The term “Point Transformer” conventionally refers to the seminal work by Zhao et al. [1], which introduced the foundational framework for point cloud processing via transformer architectures. In contrast, SplatFormer [2] represents a distinct, recently proposed extension tailored for 3D Gaussian splatting. This distinction should be explicitly clarified to avoid terminological ambiguity and ensure proper attribution.
> >
> > 2. Hierarchical Aggregation in Point-based Designs: Point Transformer [1] have demonstrated hierarchical feature learning through iterative k-nearest neighbor (k-NN) aggregation and multi-scale sampling strategies. A more rigorous discussion contrasting the proposed hierarchical mechanism with the proposed approach would strengthen the technical contribution.
> >
> > 3. Limited Validation of Framework Integration: While the integration with GS-LRM [3] is presented, the evaluation lacks generality to substantiate the method’s broader compatibility. To better demonstrate extensibility, experiments integrating with established frameworks should be included. This would more convincingly validate the architectural flexibility and scalability of the proposed approach.
> >
> > [1] Zhao H, Jiang L, Jia J, et al. Point transformer[C]//Proceedings of the IEEE/CVF international conference on computer vision. 2021: 16259-16268.
> >
> > [2] Chen Y, Mihajlovic M, Chen X, et al. Splatformer: Point transformer for robust 3d gaussian splatting[J]. arXiv preprint arXiv:2411.0
> >
> > [3] Zhang K, Bi S, Tan H, et al. Gs-lrm: Large reconstruction model for 3d gaussian splatting[C]//European Conference on Computer Vision. Cham: Springer Nature Switzerland, 2024: 1-19.
> >
> > Overall, I tend to keep my rating (Borderline reject).

---

> > > ### Author Response · Authors · 2025-08-08
> > > **Responses to the Remaining Concerns**
> > >
> > > We thank the reviewer for the detailed feedback and for clarifying the remaining concerns. Below, we address each discussion point in detail.
> > >
> > > ## **Clarification on “Point Transformer” Naming**
> > > We appreciate the reviewer for raising this valuable point. Our intention in Line 201 of the paper was to discuss the types of feed-forward 3D networks used in the relevant works, specifically grouped voxel attention in LaRA [3] and Point Transformer [1] in SplatFormer [2]. However, we acknowledge that the current wording introduces terminological ambiguity. In the revised version, we will correct the description and include a more thorough discussion of both Point Transformer and SplatFormer in the related work section. We thank the reviewer again for highlighting this issue.
> > >
> > > ## **Comparison with Hierarchical Aggregation in Point Transformer**
> > > We appreciate the reviewer’s suggestion to provide a more rigorous comparison with the hierarchical design employed in Point Transformer [1]. Local attention is adopted in Point Transformer, as applying global attention to large-scale point clouds remains challenging due to its high computational cost. As noted, Point Transformer employs a hierarchical architecture that integrates iterative k-nearest neighbor (k-NN) aggregation and multi-scale sampling. This strategy progressively expands the receptive field and facilitates multi-scale feature learning.
> > >
> > > Our method also employs hierarchical representations like Point Transformer, but adopts a fundamentally different design that enables global receptive field reasoning. Instead of using cascading local windows like Point Transformer's iterative k-NN, we introduce a coarse-to-fine voxel hierarchy and apply global attention over coarse voxels.  Our hierarchical coarse-to-fine structure with global attention directly enables long-range context aggregation while maintaining fine-grained detail and computational efficiency for large-scale scene modeling.
> > >
> > >
> > >
> > > ## **Validation of Framework Integration**
> > > We appreciate the reviewer’s suggestion to strengthen the validation of framework integration beyond GS-LRM [4]. In response, we are actively integrating our approach with a more recent baseline, DepthSplat [5]. GS-LRM and DepthSplat represent the top two existing feed-forward 3D Gaussian Splatting methods, as shown in *Table: Quantitative Comparison with Additional Baselines on the RealEstate10K Dataset* in our response to Reviewer LksN.
> > >
> > > We conducted a preliminary comparison by fine-tuning the publicly released DepthSplat checkpoint at a reduced resolution of 128×192 (original: 270×480). To further enable fast experimentation (as full-scale, full-resolution training requires more than 200 GPU hours), we used a subset of the DL3DV dataset containing 1,000 training scenes and 50 randomly selected test scenes. Both models were fine-tuned for 20,000 steps with a batch size of 8. Due to time constraints, we implemented a simplified variant of our method (Ours w/o Splat Feature in Table 1 of the paper), and it already demonstrates a notable improvement over DepthSplat, as shown in the table below.
> > >
> > >
> > > **Table: Integration of Our Method into DepthSplat**
> > >
> > > |Method|PSNR↑|SSIM↑|LPIPS↓|
> > > |---|:-:|:-:|:-:|
> > > | DepthSplat (Pretrained)           | 18.21    | 0.6980     | 0.2088     |
> > > | DepthSplat (Fine-tuned)         | 21.16    | 0.7293     | 0.1761     |
> > > | DepthSplat + Ours (Fine-tuned) | **21.79** | **0.7382** | **0.1687** |
> > >
> > > We believe this result demonstrates the extensibility and scalability of our method, and we plan to include more comprehensive experiments in the revised version to further validate its effectiveness.
> > >
> > >
> > > [1] Zhao H, Jiang L, Jia J, et al. Point transformer[C]//Proceedings of the IEEE/CVF international conference on computer vision. 2021: 16259-16268.
> > >
> > > [2] Chen Y, Mihajlovic M, Chen X, et al. Splatformer: Point transformer for robust 3d gaussian splatting[J]. arXiv preprint arXiv:2411.0
> > >
> > > [3] Chen A, Xu H, Esposito S, et al. Lara: Efficient large-baseline radiance fields[C]//European Conference on Computer Vision. Cham: Springer Nature Switzerland, 2024: 1-19.
> > >
> > > [4] Zhang K, Bi S, Tan H, et al. Gs-lrm: Large reconstruction model for 3d gaussian splatting[C]//European Conference on Computer Vision. Cham: Springer Nature Switzerland, 2024: 1-19.
> > >
> > > [5] Xu H, Peng S, Wang F, et al. Depthsplat: Connecting gaussian splatting and depth[C]//Proceedings of the Computer Vision and Pattern Recognition Conference. 2025: 16453-16463.
> > >
> > >
> > >
> > > We hope the response above helps address the reviewer’s concerns, and we would be happy to further discuss any remaining questions. We sincerely appreciate the reviewer’s thoughtful feedback and constructive engagement throughout the review process.

---

> > > ### Author Response · Authors · 2025-08-09
> > > **Official Comment by Authors**
> > >
> > > Dear Reviewer Wi3w,
> > >
> > > We would like to express our sincere gratitude for your valuable comments and the time you dedicated to the review.
> > >
> > > As the discussion period concludes **today**, we would greatly appreciate knowing whether our rebuttal has addressed the points you raised. We remain committed to clarifying any remaining questions you may have and would be happy to include additional experiments in the revised version as noted.
> > >
> > > We would be sincerely grateful if you could reconsider the score should you feel that our revisions have satisfactorily addressed your concerns. Thank you once again for your thoughtful feedback and support throughout the review process.
> > >
> > > Best regards,
> > >
> > > Authors of submission 20324

---

### Official Review · Reviewer_skmt · 2025-06-26

**Clarity:** 3
**Significance:** 3
**Originality:** 3
**Rating:** 4
**Confidence:** 4

**Summary:**

This paper proposes a Fuse-and-Refine module to enhance existing feedforward 3DGS models. The core idea is to introduce a hybrid Splat-Voxel representation as a reference for refinement, enabling efficient learning via a coarse-to-fine voxel hierarchy. The proposed method further demonstrates generalization by supporting zero-shot extension to 4DGS tasks.

**Questions:**

See the weaknesses.

**Ethical Concerns:**

["NO or VERY MINOR ethics concerns only", "Major Concern: Improper research involving human subjects"]

**Final Justification:**

The author provided detailed and somewhat persuasive responses and supplementary experiments. I agree with the idea of using 2D-3D hybrid networks for Feed-forward reconstruction in the paper. However, due to the limited improvement in performance, I maintain my score as Borderline accept.

**Limitations:**

yes.

**Paper Formatting Concerns:**

no.

**Quality:**

3

**Strengths And Weaknesses:**

Strengths:
1. The paper presents an efficient method to refine Gaussian primitives generated by existing feedforward 3DGS pipelines, improving reconstruction quality without retraining the entire system.
2. The proposed coarse-to-fine voxel hierarchy converts 3D Gaussian splats into a more structured voxel representation, which helps reduce the fitting burden on the Transformer and achieves better refinement results with improved computational efficiency.
3. The fuse idea is conceptually elegant and naturally extends to dynamic scene reconstruction, as it enables fusing primitives across time, offering promising potential for 4DGS applications.

Weaknesses:
1. While the proposed fusion strategy is conceptually helpful for mitigating redundancies caused by overlapping viewpoints, its benefit is not clearly reflected in visual quality. For example, in Figure 3 (Static scene reconstruction, bottom row), the chair’s backrest becomes somewhat transparent compared to GS-LRM. This raises concerns of hollow or missing geometry, making it unclear whether this represents a genuine improvement or a regression.
2. The method heavily relies on the quality and sparsity of the initial Gaussian primitives generated by existing methods. It seems that the method also intervenes when the pixel-aligned-based method is almost converging. Can the scheme be trained end-to-end directly?
3. The overall architecture may be similar to other voxel-based point-cloud networks. The author needs to introduce and compare these works, which are in different fields but have structural similarities.

Minors:
Figure 6: “Histoy-aware” ->“History-aware”

---

> ### Author Rebuttal · Authors · 2025-07-31
>
> We thank the reviewer for their valuable comments and appreciation of our fuse-and-refine module. Below, we address the noted weaknesses.
>
>
> ## **W1: Visual Improvement**
>
> We observe that the key visual advantage of our method over GS-LRM lies in its ability to better preserve fine details in thin structures, whereas GS-LRM often produces blurrier edges (e.g., the chair backrest case). Additional qualitative results are presented on the DL3DV dataset in Figure 9 and the RealEstate10K dataset in Figure 10, where our method consistently outperforms GS-LRM in terms of both geometric accuracy and appearance fidelity, producing sharper boundaries and more fine-grained structural details. This advantage also extends to streaming reconstruction scenarios, as illustrated in Figure 4 and the close-up examples in Figure 8.
> Quantitative metrics and visual comparisons alike further support the effectiveness of our fuse-and-refine module in enhancing reconstruction quality.
>
>
>
> ## **W2: End-to-end Training**
> We thank the reviewer for this valuable comment. Our approach supports end-to-end training together with the per-pixel primitive prediction backbone, such as the multi-view transformer used in GS-LRM. Below, we show results on the DL3DV dataset using end-to-end training from scratch for 200K iterations. We compare this to the staged training strategy in Table 1 of the paper, where the multi-view transformer is first trained alone for 100K iterations, then jointly trained with our fuse-and-refine module for another 100K iterations. To ensure a fair comparison in model capacity, both variants of our method use a 12-layer multi-view transformer, whereas GS-LRM employs a 24-layer transformer and is trained for 200K iterations.
>
> |Method|PSNR↑|SSIM↑|LPIPS↓|
> |:-:|:-:|:-:|:-:|
> |GS-LRM |28.59|0.925|0.063|
> |Ours (Staged Training)|30.34|0.934|0.054|
> |Ours (End-to-end training)|**30.61**|**0.935**|**0.052**|
>
> These results demonstrate that end-to-end training yields further improvements in performance.
>
> ## **W3: Comparison with voxel-based point-cloud networks**
>
> We thank the reviewer for the insightful suggestion regarding the connection to voxel-based point cloud networks. In the revised version, we will include a thorough discussion and citations of relevant voxel-based methods, and we would also like to highlight two key advantages of our approach over existing techniques.
>
> **Fine-grained Reconstruction in Unbounded Scene**: Voxel-based point cloud methods typically convert irregular point sets into uniform grids to enable structured operations such as 3D convolutions [1,2]. While effective for object-level 3D perception tasks, these approaches struggle to scale to large, unbounded environments. Bird’s Eye View (BEV) representations have been adopted in autonomous driving to handle large-scale point clouds more efficiently [3]. However, by projecting the scene onto a top-down plane, BEV representations inherently sacrifice vertical resolution and full 3D geometry, limiting their applicability to general-purpose 3D reconstruction. In contrast, our method constructs a cost volume aligned with the target camera's normalized device coordinates (NDC), where the z-axis is parameterized linearly in disparity to enable finer modeling of near-field geometry. This voxel representation supports high-fidelity reconstruction across large, unconstrained scenes.
>
> **​​Global Perception Field:** While attention mechanisms have been introduced to voxel-based point cloud networks to overcome the locality constraints of traditional 3D convolutions, most approaches [4,5] are still limited to local attention due to memory constraints. Full global attention over all voxels or points remains challenging especially for large-scale unbounded scenes. We address this challenge by introducing a coarse-to-fine voxel hierarchy that enables global attention at the coarsest level. This design facilitates efficient long-range context aggregation while capturing fine-grained details essential for large-scale scene modeling.
>
> [1]: Zhou Y, Tuzel O. Voxelnet: End-to-end learning for point cloud based 3d object detection
>
> [2]: Liu Z, Tang H, Lin Y, et al. Point-voxel cnn for efficient 3d deep learning
>
> [3]: Li Z, Wang W, Li H, et al. Bevformer: learning bird's-eye-view representation from lidar-camera via spatiotemporal transformers
>
> [4]: Mao J, Xue Y, Niu M, et al. Voxel transformer for 3d object detection
>
> [5]: He C, Li R, Li S, et al. Voxel set transformer: A set-to-set approach to 3d object detection from point clouds

---

> > ### Comment · Reviewer_skmt · 2025-08-05
> >
> > I appreciate the author's detailed response, especially the clarification on end-to-end training, which prompts me to have another question: Can the module proposed by the method be trained independently as a reconstruction network, that is, completely independent of pixel-aligned GS generation?

---

> > > ### Author Response · Authors · 2025-08-05
> > > **Official Comment by Authors**
> > >
> > > We thank the reviewer for their thoughtful response and follow-up question.
> > >
> > > Our voxel transformer can indeed function as a standalone reconstruction network without relying on pixel-aligned Gaussian Splatting. Instead of performing the splat-to-voxel transfer as described in the paper, we can directly lift 2D image features into a volumetric representation. The voxel transformer then follows the same procedure to process this volume and generate per-voxel Gaussians. A relevant work, LaRa: Efficient Large-Baseline Radiance Fields, explores a similar idea and demonstrates its effectiveness. While LaRa employs a grouped attention mechanism to process the 3D volume, our voxel transformer adopts a coarse-to-fine hierarchical design which offers the potential to reconstruct finer-grained details with higher quality.
> > >
> > > We would be happy to clarify further if needed, and would be sincerely grateful if you would consider raising the score in support of our method. Thank you once again for your valuable feedback and support throughout the review process.

---

> > > > ### Comment · Reviewer_skmt · 2025-08-06
> > > >
> > > > I appreciate the author's quick response. I understand that due to time constraints, it may not be possible to add experiments for this part of the additional questions during this period. I will reconsider the score after the AC-reviewer discussion.

---

> > > > > ### Author Response · Authors · 2025-08-06
> > > > > **Official Comment by Authors**
> > > > >
> > > > > Thank you very much for your kind understanding and for taking the time to engage thoughtfully with our responses. We sincerely appreciate your support, your consideration, and your willingness to revisit the score after the AC-reviewer discussion. We would also be glad to include the additional experiments in the revised version to further strengthen the contribution.

---

### Official Review · Reviewer_LksN · 2025-07-01

**Clarity:** 3
**Significance:** 3
**Originality:** 3
**Rating:** 5
**Confidence:** 4

**Summary:**

The paper presents a novel method for efficient 3D scene reconstruction using a Fuse-and-Refine module that enhances feed-forward 3D Gaussian Splatting. It shows several applications both in static and dynamic reconstructions.

The Fuse-and-Refine module is motivated by the fact that existing GS LRM models typically predict a gsplat at every incoming pixel, which is very redundant and won't scale well to long sequence or dynamic scenes.
It uses a multi-scale (two scale) voxel structure to "splat" the splats, prune the undersampled, then globally refine the voxels on the coarse-to-fine level through a transformer, then finally decode the voxels back to gsplats.
This module can be trained *after* a pretrained GS-LRM and will quickly converge achieving higher visual quality. The module is also designed with performance in mind, achieving higher PSNR compared to pixel-aligned GS-LRM with similar runtime.

**Questions:**

The online splat-voxel construction should ideally enable large scene online reconstruction.
I'm curious how this paper would compare with other splat-voxel representations such as
- SCube: Instant Large-Scale Scene Reconstruction using VoxSplats (code available)
- Omni-Scene: Omni-Gaussian Representation for Ego-Centric Sparse-View Scene Reconstruction

In table 4, does the reported time includes also tracking and 3d warping? Is it possible to report each step individually?
In the same table, I'm not sure why 3dgs rime is fast? is this the training time or rendering time? in appendix you mentioned shortening the training steps significantly, while the psnr on new views grow 4db, it is not a good reason to use not converged 3dgs.

**Ethical Concerns:**

["NO or VERY MINOR ethics concerns only"]

**Final Justification:**

My concern over dynamic scene reconstruction wasn’t raised by other reviewers, I concede that the emphasis on that is reasonable for the online streaming reconstruction as the author pointed out.

**Limitations:**

I think it has some really good ideas and should work well on long sequence static scene reconstruction.
I reservation is rooted in the paper's strong claim in dynamic scenes. While it is impressive that the method can handle dynamic scenes
without training on dynamics, its dynamic expression mainly comes from continuous tracking (via existing 2D tracking) and robust fusion. From the presented video, such robust fusion often translates to visual "averaging",  in terms of reconstruction quality, it will never reach the level of a model that actually trains with dynamic scenes.

The comparison on this topic focuses on rather simple sequences. Furthermore, the comparison is done with methods that don't support dynamic reconstruction natively. I'm not convinced of the application.

Though I do like the methodology, and think it could be a great paper if the authors focused only on static scenes and built a compelling case in static scene reconstruction by comparing with other strong baselines, such as SCube.

**Paper Formatting Concerns:**

No concerns.

**Quality:**

3

**Strengths And Weaknesses:**

# Strength
- High impact: breaking the old format of per-pixel gsplat is a key to scalability, this paper address this challenge in a refreshing way, which is worth sharing to the community.
- Practical application: high throughput for streaming application, while significantly reduce the gaussian counts, hence further improve rendering speed.
- Well written, well structured experiments and ablation studies.

# Weakness
- The position of the paper is not very convincing to me. While spending significantly amount of time on building a use case for dynamic scene reconstruction, the method is ultimately still for static-scene reconstruction. In the video, the model clearly struggles with highly dynamic components, such as the flame. I think being able to robust handle inconsistent input views is a benefit of voxel and global refinement, but it shouldn't lead to claims of 4D scene reconstruction.
- Related, I think the comparison for novel-view streaming is not really fair. 3DGS and GS-LRM doesn't have any dynamic component there, equivalent 3D warping should be added in order for the comparison to make sense.

---

> ### Author Rebuttal · Authors · 2025-07-31
>
> We thank the reviewer for recognizing our fuse-and-refine strategy, as well as for the constructive feedback regarding dynamic scene reconstruction and the suggestion to place greater emphasis on static scenes. Below, we address the discussion points in detail.
>
>
>
> ## **W1&W2&Limitation: Dynamic Scene Reconstruction**
> We would like to clarify that our application to dynamic scene reconstruction specifically focuses on sparse-view online streaming reconstruction, a particularly challenging and underexplored setting within the field. By achieving interactive reconstruction rates of 15 FPS from sparse-view, long streaming input, our method becomes highly applicable to online augmented and virtual reality scenarios. Existing methods fall short in this scenario: optimization-based methods such as 3DGS and 4DGS struggle with sparse-view inputs, face scalability issues with long video sequences, and fail to meet the reconstruction speed requirements of online applications; feed-forward approaches like GS-LRM do not leverage historical information to ensure temporal consistency, often leading to noticeable flickering artifacts.
>
> Thanks to the reviewer's suggestion, we additionally apply 2D tracking and 3D warping to GS-LRM and present the results below.  We use the same keyframing strategy as described in the paper, refreshing the warped primitives every 5 frames to mitigate error accumulation.
>
>
> **Table: Comparison for Adding 3D Warping to GS-LRM**
>
> |Method|PSNR↑|SSIM↑|LPIPS↓|Flicker↓|
> |---|:-:|:-:|:-:|:-:|
> |GS-LRM|21.80|0.8488|0.1278|5.714|
> |GS-LRM +  3D Warping|21.40|0.8250|0.1419|**2.163**|
> |GS-LRM + 3D Warping + Fusion (Concat+Dropout)|19.52|0.7429|0.2502|3.983|
> |GS-LRM + 3D Warping + Ours (Fuse-and-Refine)|**27.41**|**0.8863**|**0.1040**|2.916|
>
>
> The results highlight that leveraging historical information by **directly applying 3D warping to per-frame reconstruction backbones is prone to error**. Specifically, warping past reconstructions to later frames (GS-LRM + 3D Warping) leads to larger reconstruction errors, as it fails to handle newly appearing content and suffers from accumulated warping errors.  Addressing these limitations is non-trivial. A simple fusion baseline (Concat + Dropout) incrementally adds new primitives while discarding 50% per frame to prevent unbounded growth in the number of primitives, but lacks refinement and often removes accurate primitives.
>
>
> To overcome these challenges, we propose a novel fuse-and-refine mechanism. The module efficiently leverages historical information, functions at interactive rates, and does not require training on the currently limited multi-view temporal datasets. We believe this robust fusion strategy provides a compelling and practical solution to the challenges of online streaming reconstruction.
>
> We also acknowledge the current limitations in performance, for example in the dynamic components as noted by the reviewer. We believe these challenges are due in part to our simple 3D warping strategy, which may have limited accuracy under the sparse-view setting. Future work could explore more advanced 3D warping backbones to address this issue, which is beyond the scope of this work. Due to the scarcity of available multi-view video data, we currently train our method only on static scenes. Nonetheless, the fuse-and-refine framework holds strong potential for training on dynamic scenes, which could further mitigate temporal artifacts in future work.
>
>
>
>
> ## **Limitation: Extended Evaluation for Static Scene Reconstruction**
>
> We appreciate the reviewer’s recognition of our method’s potential for static scene reconstruction. As shown below, our method outperforms recent feed-forward 3D Gaussian Splatting methods on RealEstate10K:
>
> **Table: Quantitative Comparison with Additional Baselines on the RealEstate10K Dataset**
> |Method|PSNR↑|SSIM↑|LPIPS↓|
> |:----:|:---:|:---:|:-----:|
> |pixelSplat [CVPR2024]|26.09|0.863|0.136|
> |MVSplat [ECCV2024]|26.39|0.869|0.128|
> |GS-LRM [ECCV2024]|28.10|0.892|0.114|
> |TranSplat [AAAI2025]|26.69|0.875|0.125|
> |HiSplat [ICLR2025]|27.21|0.881|0.117|
> |Omni-Scene [CVPR2025]|26.19|0.865|0.131|
> |DepthSplat [CVPR2025]|27.47|0.889|0.114|
> |Ours|**28.47**|**0.907**|**0.078**|
>
>
>
> Our method also demonstrates strong generalization capability across varying input views. We compare it with Gaussian Graph Network, a state-of-the-art method for handling diverse input views through local primitive fusion using graph convolution. We follow its experimental setup to train the model with 2 input views and evaluate with 4 input views on the RealEstate10K dataset. As shown in the table below, our method not only significantly outperforms the Gaussian Graph Network but also maintains consistent reconstruction quality when generalizing from 2 to 4 input views.
>
>
> **Table: Comparison with Gaussian Graph Network on the RealEstate10K Dataset (2-view training, 4-view evaluation)**
>
> |Method|PSNR↑|SSIM↑|LPIPS↓|
> |:-:|:-:|:-:|:-:|
> |Gaussian Graph Network|24.76|0.784|0.172|
> |**Ours**|**28.79**|**0.914**|**0.081**|
>
>
>
> Our method also performs well on large unbounded scene datasets such as DL3DV. As shown in paper Table 1 and Appendix Figure 9, our fuse-and-refine approach significantly improves the reconstruction quality of GS-LRM, yielding a PSNR improvement of approximately 2dB while maintaining a comparable model size.
>
> Trained with 4 input views on the DL3DV dataset, our method consistently outperforms GS-LRM across a wide range of input configurations as shown in the table below. Since input views are chosen based on proximity to the target, increasing from 4 to 16 introduces a distribution shift while contributing limited new information for target view rendering, which explains the performance drop in both methods. Nevertheless,  our method maintains significantly better reconstruction quality than GS-LRM across all input settings. We believe these results further demonstrate the potential of our method to handle long sequence large scene reconstruction, as noted by the reviewer.
>
> **Table: Quantitative Comparison under Varying Numbers of Input Views on the DL3DV Dataset (4-view Training)**
>
>
> |Method||2 views|||4 views|||8 views|||16 views||
> |:----:|:-----:|:--:|:------:|:-----:|:-----:|:------:|:-----:|:-----:|:------:|:------:|:------:|:--------:|
> |      |PSNR↑|SSIM↑|LPIPS↓|PSNR↑|SSIM↑|LPIPS↓|PSNR↑|SSIM↑|LPIPS↓|PSNR↑|SSIM↑|LPIPS↓|
> |GS-LRM|18.99|0.810|0.135|28.59|0.925|0.063|22.51|0.879|0.106|20.12|0.792|0.177|
> |Ours|**26.32**|**0.876**|**0.093**|**30.34**|**0.934**|**0.054**|**28.98**|**0.930**|**0.060**|**26.06**|**0.889**|**0.093**|
>
>
> We would also be happy to include more evaluations on static scene reconstruction, and would appreciate it if the reviewer could suggest specific experimental setups.
>
>
>
>
> ## **Q1: Comparison with Splat-voxel Representations**
> We thank the reviewer for highlighting relevant voxel-splat representations and will include the following discussion in the revised related works section.
> **SCube** pretrains a voxel diffusion model using LiDAR and COLMAP point clouds from a self-driving dataset to infer voxel representations at inference. However, it is extremely slow (several seconds per frame) and not applicable to scenes like RealEstate10K that lack LiDAR input. In contrast, our method is trained using only RGB supervision. It also supports large-scale scene reconstruction such as DL3DV, and achieves significantly faster inference times of approximately 70 milliseconds per frame. **Omni-Scene** combines pixel-aligned Gaussians with Gaussians constructed from feature volumes, aiming to exploit the strengths of both pixel-based and volumetric representations. Unlike our method, this design is not easily extendable to dynamic scenes. Additionally, both SCube and Omni-Scene rely on 3D CNNs to process voxel features in a tractable manner, but this design leads to limited perceptual fields. In contrast, our approach introduces an efficient coarse-to-fine voxel hierarchy, allowing us to directly perform full self-attention over the input tokens. As shown in the comparison on the RealEstate10K dataset above, our method significantly outperforms Omni-Scene and achieves state-of-the-art performance.
>
>
>
> ## **Q2: Runtime Breakdown**
>
> The runtime reported in paper Table 4 includes both 3D warping and 2D tracking times. 2D tracking is applied only to keyframes, sampled every 5 frames. Below, we provide a detailed runtime breakdown for both streaming and static scene reconstruction (tested on H100).
>
>
> **Table: Runtime Breakdown of Streaming Reconstruction on the Neural3DVideo Dataset**
>
> |Components|Multi-view Transformer|2D Tracking|3D Warping|Temporal Fuse-and-Refine|Rendering|
> |:-:|:-:|:-:|:-:|:-:|:-:|
> |Time(s)|0.030|0.28|0.011|0.026|0.003|
>
> **Table: Runtime Breakdown of Static Scene Reconstruction on the RealEstate10K Dataset**
> |Components|Multi-view Transformer|Splat-to-Voxel Transfer|Voxel Transformer|Rendering|
> |:-:|:-:|:-:|:-:|:-:|
> |Time(s)|0.026|0.005|0.016|0.003|
>
>
> ## **Q3: Performance of Converged 3DGS**
> The runtimes of 3DGS and 3DGStream reported in paper Table 4 include both training and rendering times. We apply early termination during training due to a severe overfitting problem observed in 3DGS under the sparse-view setup. To illustrate this, we show the training and validation PSNR curves for 3DGS on a single frame from the 2-view Neural 3D Video dataset.
>
> |Training Iterations|500|1000|2000|3000|10000|15000|
> |:-:|:-:|:-:|:-:|:-:|:-:|:-:|
> |Training Time(s)|4|9|15|22|72|108|
> |Training PSNR|16.84|17.91|37.25|45.58|48.34|**50.11**|
> |Validation PSNR|16.88|**20.88**|17.72|17.52|13.64|13.39|
>
>
> As shown in the table, validation PSNR peaks near 1000 iterations while training PSNR continues to rise, clearly indicating overfitting. This observation motivates our decision to limit training iterations. However, we are also happy to include the fully converged 3DGS and 3DGStream results using their original configurations in the revised version.

---

### Official Review · Reviewer_8DB3 · 2025-07-03

**Clarity:** 3
**Significance:** 2
**Originality:** 2
**Rating:** 5
**Confidence:** 4

**Summary:**

The paper lies in the space of feed-forward 3D Gaussian Splatting (3DGS) and is built on top of a feed-forward 3DGS model (GS-LRM) which predicts 3D Gaussian primitives from sparse-view images. The first contribution is a Splat-to-Voxel module, which aims to reduce redundancy and improve density control by assigning the 3D Gaussian primitives to fine and coarse voxel grids and fusing them in a coarse-to-fine manner. The second contribution is a Sparse Voxel Transformer, which enables learning refined Gaussian parameters from coarse voxel features. The Sparse Voxel Transformer is trained on static scenes, but it is shown that it can also generalise to dynamic scenes, where the proposed fuse-and-refine strategy is used to improve temporal consistency in streaming scene reconstruction.

**Questions:**

1. Does voxel sparsification effectively reduce the number of Gaussian primitives at the output of the feed-forward model? If so, is removing entire (coarse) voxels the most effective approach to culling? Wouldn’t culling at the level of primitives or even fine voxels provide better density control, without creating “dead” regions? Why or why not?
2. How is the 20% threshold in voxel sparsification decided?  An ablation here would also be helpful.
3. Can the Voxel Transformer be applied at the fine voxel level, and how would that affect performance?
4. It is mentioned in the abstract that the model can process **20k** primitives in 15ms, and in L189 that the voxel transformer processes fewer than **10k** tokens in 15ms. Can the authors explain this further?
5. How are the keyframes chosen in the streaming setting?
6. In Table 4, why do the keyframe and non-keyframe variants give the same result?
7. Can the streaming fusion be applied to monocular videos?

**Ethical Concerns:**

["NO or VERY MINOR ethics concerns only"]

**Final Justification:**

The rebuttal has addressed my open questions well, and I am happy to increase my rating. Reading the other reviewers and responses, I did not find any strong concerns that would prevent acceptance.

**Limitations:**

Limitations are sufficiently discussed in the paper, but some are primarily attributed to the underlying methods that the paper builds upon (such as limitations of the feed-forward model GS-LRM). I would have also liked to see a discussion of potential limitations of the approach itself.

**Paper Formatting Concerns:**

There are no major formatting concerns.

A minor note: Tables 3 and 4 seem to exceed the page width.

**Quality:**

3

**Strengths And Weaknesses:**

## Strengths

* The paper has a clear structure and is well-written, making it easy to understand.
* The paper improves reconstruction performance over the baseline model (GS-LRM) with an efficient fuse-and-refine strategy, which takes only 15ms. This strategy reduces redundancy (through voxelization) and refines the Gaussian primitives by leveraging global context (through coarse-voxel features).
* While the approach is trained on static scenes only, the paper also evaluates on a dynamic dataset, demonstrating the generalisation ability of the model to streaming scene reconstruction. I find this result particularly interesting, even if it may lag behind tailored dynamic scene reconstruction solutions.

## Weaknesses

A major weakness of the paper is the lack of a detailed discussion and comparisons to related work [1,2,3,4,5]. In particular, quantitative comparisons could include more recent methods such as TranSplat [1] and EVolSplat [2] in the static setting, where the latter also focuses on efficiency, or HiCoM [5] in the dynamic setting. More importantly, Gaussian Graph Network [3] and HiSplat [4] are closely related to this paper, aiming to aggregate Gaussian primitives with graph operations or fusion modules. To fully judge the contribution and effectiveness of the paper, a thorough discussion and, where possible, comparisons to these works should be provided.

Due to the focus on efficiency, the paper should provide a more detailed breakdown of the computation times (e.g., feedforward model, splat-to-voxel transfer, voxel transformer, rendering)  and a time comparison to prior work on the static scene setting. It would be interesting to see in more detail how the voxel sparsification affects efficiency.

Some more model considerations / ablations could be discussed or included in the paper (see Questions).

### References
[1] Zhang, Chuanrui, et al. "Transplat: Generalizable 3d gaussian splatting from sparse multi-view images with transformers." AAAI 2025.

[2] Miao, Sheng, et al. "Evolsplat: Efficient volume-based gaussian splatting for urban view synthesis." CVPR 2025.

[3] Zhang, Shengjun, et al. "Gaussian graph network: Learning efficient and generalizable gaussian representations from multi-view images." NeurIPS 2024.

[4] Tang, Shengji, et al. "Hisplat: Hierarchical 3d gaussian splatting for generalizable sparse-view reconstruction." ICLR 2025.

[5] Gao, Qiankun, et al. "Hicom: Hierarchical coherent motion for dynamic streamable scenes with 3d gaussian splatting." NeurIPS 2024.

---

> ### Author Rebuttal · Authors · 2025-07-30
>
> We appreciate the reviewer’s valuable feedback and their recognition of our method, especially its ability to generalize to streaming scene reconstruction. In the following, we address the identified weaknesses in detail.
>
> ## **W1: Discussion and Comparison with Related Work**
> We thank the reviewer for highlighting relevant works for comparison and discussion, which we will thoroughly incorporate into the revised version. Here, we outline the key advantages of our method over **HiSplat** and **Gaussian Graph Network**, both of which also explore aggregation strategies for Gaussian primitives.
>
> **HiSplat** predicts multi-scale per-pixel Gaussian primitives and fuses these hierarchical predictions at each pixel location. However, the absence of a 3D canonical space for fusion limits its ability to generalize to temporal fusion of Gaussian primitives across frames, which our method is explicitly designed to support. Furthermore, our sparse voxel transformer learns an effective global strategy using full attention. This results in substantially improved performance on the RealEstate10K dataset (shown in the table below) compared to the local per-pixel fusion approach in HiSplat.
> **Gaussian Graph Network** fuses per-pixel primitives using graph convolutions over local neighborhoods, but its scalability is limited since the number of edges grows quadratically with the number of input primitives. By transferring an arbitrary number of primitives into a fixed voxel representation, our voxel transformer maintains constant processing time regardless of input size. Moreover, graph convolutions are inherently limited by their local receptive fields, making it difficult to capture the global organization of primitives. Our approach overcomes these limitations by leveraging full attention within the voxel transformer to aggregate information across the entire scene, enabling long-range context modeling and global structural fusion and refinement.
>
> We also include a quantitative comparison with more recent feed-forward 3D Gaussian Splatting methods including **TranSplat**, **HiSplat**, **Omni-Scene** on the RealEstate10K dataset. As shown below, our method achieves the best performance among all baselines, demonstrating its effectiveness. Since **EvoSplat** does not report results on RealEstate10K, we include **Omni-Scene** instead, a related method that similarly adopts a voxel-splat representation for enhancing driving scenarios.
>
>
> **Table: Additional Baseline Comparisons on the RealEstate10K Dataset**
>
> |Method|PSNR↑|SSIM↑|LPIPS↓|
> |:-:|:-:|:-:|:-:|
> |Omni-Scene|26.19|0.865|0.131|
> |TranSplat|26.69|0.875|0.125|
> |HiSplat|27.21|0.881|0.117|
> |**Ours**|**28.47**|**0.907**|**0.078**|
>
>
>
>
> We also compare our method with **Gaussian Graph Network** on the RealEstate10K dataset, following its experimental configuration. Both models are trained with 2 input views and evaluated with 4 input views to assess generalization across varying input configurations. As shown in the table below, our method significantly outperforms this state-of-the-art Gaussian fusion approach.
>
> **Table: Comparison with Gaussian Graph Network on the RealEstate10K Dataset (2-view training, 4-view evaluation)**
>
> |Method|PSNR ↑|SSIM ↑|LPIPS ↓|
> |:-:|:-:|:-:|:-:|
> |Gaussian Graph Network|24.76|0.784|0.172|
> |**Ours**|**28.79**|**0.914**|**0.081**|
>
>
>
>
> Due to time constraints, we could not include recent streaming methods like HiCoM in the rebuttal, but we will add them in the revision. *HiCoM*, like 3DGStream, is designed for dense-view input and performs poorly under sparse views. While faster than 3DGStream, it still takes over 1 second per frame, making it unsuitable for online reconstruction. In contrast, our method runs at 0.07 seconds per frame.
>
>
>
> ## **W2: Efficiency Analysis**
>
> We present a detailed breakdown of the inference runtime of our method, evaluated on an NVIDIA H100 GPU. We first report the runtime breakdown for static scene reconstruction on RealEstate10K using two input views at 256×256 resolution.
>
> **Table: Runtime Breakdown of Static Scene Reconstruction on the RealEstate10K Dataset**
>
> |Components|Multi-view Transformer|Splat-to-Voxel Transfer|Voxel Transformer|Rendering|
> |:-:|:-:|:-:|:-:|:-:|
> |Time(s)|0.026|0.005|0.016|0.003|
>
>
>
> We further analyze the runtime of each component in a streaming reconstruction setting on the Neural3DVideo dataset, using 2 input views at a resolution of 320×240. Note that 2D tracking is applied only to keyframes, which are sampled every 5 frames.
>
>
>
> **Table: Runtime Breakdown of Streaming Reconstruction on the Neural3DVideo Dataset**
>
> |Components|Multi-view Transformer|2D Tracking|3D Warping|Temporal Fuse-and-Refine|Rendering|
> |:-:|:-:|:-:|:-:|:-:|:-:|
> |Time(s)|0.030|0.28|0.011|0.026|0.003|
>
>
>
> We also present an inference time comparison between our method and other feed-forward Gaussian Splatting approaches, including PixelSplat, MVSplat, and GS-LRM, on the RealEstate10K dataset. All runtimes were measured on an NVIDIA A100 GPU.
>
> **Table: Inference Time Comparison on the RealEstate10K Dataset**
>
> |Method|PSNR↑|SSIM↑|LPIPS↓|Inference Time(s)|
> |:-:|:-:|:-:|:-:|:-:|
> |pixelSplat|26.09|0.863|0.136|0.104|
> |MVSplat|26.39|0.869|0.128|0.044|
> |GS-LRM|28.10|0.892|0.114|**0.041**|
> |**Ours**|**28.47**|**0.907**|**0.078**|0.067|
>
> The slight runtime overhead of our method compared to GS-LRM can be mitigated by reducing the number of multi-view transformer layers. As shown in Table 1 of our paper, our method improves PSNR by over 2 dB while maintaining comparable model size and inference time.
>
>
> ## **Q1&Q3: Effect of Voxel Sparsification and Applying Transformer on Fine-level voxels**
> The primary goal of voxel sparsification is to reduce the number of input tokens for the voxel transformer. Without sparsification, the token count would scale cubically with the voxel dimensions, with excessive GPU memory usage and slow compute speed. Relatedly, applying the transformer on coarse-level voxels also helps to reduce token count, whereas directly applying the transformer on fine-level voxels becomes infeasible unless we reduce voxel resolution which harms reconstruction quality. While the reviewer is correct that culling the fine voxels can improve rendering speed by reducing the final number of primitives, our coarse-level sparsification achieves two goals: reduceing the final primitive count and making the voxel transformer tractable.
>
> As shown in Table 1, we include ablation studies for the setting without voxel sparsification (Ours (w/o Coarse-to-Fine)). Due to memory constraints, we reduce the voxel resolution by 75% in this setting, which still results in slower inference and degraded reconstruction quality. Similarly, in the variant without the coarse-to-fine hierarchy (Ours (w/o Coarse-to-Fine)), the voxel transformer is applied only at a single voxel level. Applying the transformer directly on dense-level voxels would require an even greater reduction in resolution due to memory limitations, leading to significantly worse results.
>
>
> ## **Q2&Q4: Sparsification Ratio and Token/Primitive Numbers**
> We apologize for the typo in the abstract regarding the number of primitives. The correct number is 200K (as stated in line 57).
> The number of primitives and tokens is calculated as follows: given 2 to 4 input views with a resolution of 384 × 216, GS-LRM generates multi-view per-pixel primitives (2~4 × 384 × 216), resulting in approximately 165K to 330K primitives. For these primitives, Our splat-to-voxel transfer converts them into a coarse voxel grid with a resolution of (384 / 8) × (216 / 8) × 32. We then sparsify the grid based on voxel opacity and retain only the top 20%, yielding roughly 8K tokens for the transformer input. This voxel transformer takes approximately 15 milliseconds for the forward pass, as shown in the breakdown above.
>
> The sparsification ratio (20% threshold) is chosen to keep the number of tokens below 10K, ensuring that the transformer runs within 15 milliseconds. Although using a higher sparsification ratio can further improve speed, it risks discarding useful information. Conversely, a lower ratio increases token count and slows down inference. In practice, we find that a sparsification ratio between 15% and 25% offers a good balance between performance and quality. We also observed that sparsity is common in large-scale unbounded scene modeling, with typically fewer than 30% of voxels having a primitive opacity greater than 0.005.
>
>
> ## **Q5&Q6: Keyframing**
> We select a keyframe every 5 frames to run the 2D tracking model, which takes approximately 0.28 seconds and provides tracking results for the following 5 frames. Only two keyframes are maintained during streaming reconstruction to ensure efficiency.
>
> As expected, non-keyframe frames may exhibit slightly reduced performance compared to keyframes due to accumulated 3D warping errors. However, we report the average performance over the entire sequence in paper Table 4, which reflects the practical effectiveness of our method under the streaming setting.
>
> ## **Q7: Monocular Input**
> Our method currently does not support monocular streaming, as the GS-LRM backbone requires at least two input views to produce reliable Gaussian primitives. However, with a reliable single-view Gaussian Splatting predictor, our framework could naturally be extended to support monocular reconstruction.
>
> ## **Limitation**
> We acknowledge that coarse-level pruning may introduce “dead regions” in the reconstruction. However, we observe that typical 3D scenes do contain large regions of empty space, and our pruning step exploits this fact to improve processing efficiency. We currently use a simple heuristic, assuming a fixed sparsity ratio of 20%, but a potential direction for future work is a more adaptive and scene-aware pruning strategy.

---

> > ### Author Response · Authors · 2025-08-05
> > **Official Comment by Authors**
> >
> > Dear Reviewer 8DB3,
> >
> > We would like to sincerely thank you for your valuable feedback and the time you’ve dedicated to reviewing our paper.
> >
> > As the discussion phase approaches its end, we hope our responses have addressed your concerns. If there are any remaining issues or questions, we would be happy to clarify further. If you find our revisions satisfactory, we would be truly grateful if you would consider raising your score for our paper.
> >
> > Thank you once again for your thoughtful comments and support throughout the review process.

---

> > > ### Comment · Reviewer_8DB3 · 2025-08-06
> > >
> > > Thank you for the response, clarifications, and additional experiments. All my concerns have been addressed, and I do not find any major negative points in the other reviews, so I will increase my rating.

---

> > > > ### Author Response · Authors · 2025-08-09
> > > > **Thank you for your support**
> > > >
> > > > We’re very glad that all your concerns have been adequately addressed, and we’re sincerely grateful for your increased rating and support.

---

### Note · Authors · 2025-08-16

We sincerely thank all reviewers and the AC for their constructive feedback and productive discussion. We are encouraged by the recognition that our proposed fuse-and-refine method is both novel and effective. In our rebuttal, we strengthened the paper by addressing several key issues raised during the insightful exchanges with the reviewers:


1. **Expanded Discussion of Related Work**

We broadened our discussion to include recent feed-forward 3DGS methods (Reviewer 8DB3), splat–voxel representations (Reviewer  LksN), voxel-based point cloud networks (Reviewer  skmt), and SplatFormer (Reviewer Wi3w).  This clarifies the novelty and advantages of our approach and strengthens the paper’s technical positioning.

2. **Extended Evaluation**

*Static scenes*: We included comprehensive comparisons with recent feed-forward 3DGS methods on *RealEstate10K*, demonstrating state-of-the-art performance over all existing approaches. We further demonstrated strong generalization across varying input views, highlighting the potential of our method for long-sequence, large-scale scene reconstruction. We also demonstrated extensibility and scalability by integrating our approach into another leading feed-forward GS method, *DepthSplat* (in addition to *GS-LRM*).

*Streaming reconstruction*: We established stronger baselines by augmenting *GS-LRM* with 3D warping and simple fusion strategies, demonstrating the advantage and necessity of our proposed temporal fusion module.

3. **Efficiency Analysis**

We present a detailed breakdown of the inference runtime for each component in both static and streaming reconstruction. We also provide runtime comparisons with other feed-forward 3DGS methods. Our approach delivers state-of-the-art performance in both static and streaming scene reconstruction while running at interactive rates. It can achieve notable improvements compared to baselines, while keeping model size and inference time comparable (Table 1).



We are pleased that the concerns of three reviewers have been addressed, and we have provided additional experiments and responses to resolve Reviewer Wi3w’s remaining point. Overall, the constructive feedback from all reviewers has been invaluable in refining our work, and we will incorporate the suggested points and promised revisions into the final version.

---

### Decision · Program_Chairs · 2025-09-17

**Decision:**

Accept (poster)

**Comment:**

This paper considers the task of feed-forward 3D Gaussian Splatting (3DGS) from images, where pixelwise Gaussians are processed in 3D space to reduce redundancy, control density and refine the primitives with global context. The identified strengths include that the approach processes Gaussians in 3D-space rather than image-space, facilitating merging and refinement; the reconstruction performance; the good balance between fidelity and throughput; the unexpected and interesting generalization to streaming scene reconstruction; and the structure and clarity of the writing. The identified weaknesses include a significant amount of closely related work that was not discussed in the submission [8DB3, LksN, skmt, Wi3w]; closely related work that was not quantitatively compared to the proposed approach [8DB3, LksN]; missing efficiency analyses [Wi3w]; concerns about the dynamic scene performance and analysis [skmt, Wi3w]; and compatibility with other GS frameworks [Wi3w]. Several of these concerns were addressed in the author responses to the satisfaction of the reviewers, including the literature discussion, comparisons, efficiency analyses, and the dynamic streaming setting.

The initial ratings were relatively consistent but borderline (2xBR/2xBA). After reading all reviews, author responses, and engaging in discussion with the authors and each other, the reviewers are on-the-whole positive about this work, with the modal response recommending acceptance. The dissenting reviewer has ongoing concerns with the technical differences w.r.t. SplatFormer and the compatibility of the method with other frameworks. The latter, in the AC's view, was satisfactorily addressed in the authors final response, and the former can be handled by extending the discussion about this work in the revision. Overall, the AC sees no reason to override the view of the reviewer majority and is satisfied by the technical merit of the paper and its potential to make an impact on the field.